# A Review of Cytotoxic Plants of the Indian Subcontinent and a Broad-Spectrum Analysis of Their Bioactive Compounds

**DOI:** 10.3390/molecules25081904

**Published:** 2020-04-20

**Authors:** Kishor Mazumder, Biswajit Biswas, Iqbal Mahmud Raja, Koichi Fukase

**Affiliations:** 1Department of Pharmacy, Jashore University of Science and Technology, Jashore 7408, Bangladesh; bb@just.edu.bd (B.B.); rajagoja95@gmail.com (I.M.R.); 2School of Biomedical Sciences, Charles Sturt University, Boorooma St, Locked Bag 588, Wagga Wagga, New South Wales 2678, Australia; 3Department of Chemistry, Graduate School of Science, Osaka University, Osaka 565-0871, Japan

**Keywords:** anticancer, cytotoxic, Indian subcontinent plants, bioactive compounds

## Abstract

Cancer or uncontrolled cell proliferation is a major health issue worldwide and is the second leading cause of deaths globally. The high mortality rate and toxicity associated with cancer chemotherapy or radiation therapy have encouraged the investigation of complementary and alternative treatment methods, such as plant-based drugs. Moreover, over 60% of the anti-cancer drugs are molecules derived from plants or their synthetic derivatives. Therefore, in the present review, an attempt has been made to summarize the cytotoxic plants available in the Indian subcontinent along with a description of their bio-active components. The review covers 99 plants of 57 families as well as over 110 isolated bioactive cytotoxic compounds, amongst which at least 20 are new compounds. Among the reported phytoconstituents, artemisinin, lupeol, curcumin, and quercetin are under clinical trials, while brazilin, catechin, ursolic acid, β-sitosterol, and myricetin are under pharmacokinetic development. However, for the remaining compounds, there is little or no information available. Therefore, further investigations are warranted on these subcontinent medicinal plants as an important source of novel cytotoxic agents.

## 1. Introduction

Cancer is a severe metabolic disorder and the leading cause of death worldwide [1]. It involves unrestrained proliferation of normal cells, caused by genetic alterations and instabilities, resulting in the generation of malignant cells and initiation of metastasis or tissue invasion. The genetic alterations include mutation of tumor suppressor genes (*NF1*, *NF2*, *p53* etc.), DNA repair genes (tool box for DNA and *p21*, *p22*, *p27*, *p51*, *p53*), oncogenes (*MYC*, *RAS*, *Bcl-2*, *RAF*), and genes involved in cell growth and metabolism [2]. These mutations are caused by internal factors, such as alteration in hormonal balance and immune system as well as by external factors, such as radiation and pesticide exposure, tobacco smoking, and ingestion of carcinogenic chemicals or metals [1]. The incidence and geographic distribution of cancer are also related to parameters such as gender, age, race, genetic predisposition, and exposure to environmental carcinogens including azo dyes, aflatoxins, petrol, and mutagenic agents [3].

Since time immemorial, drugs of natural origin have formed the basis of traditional medicine in different cultures and in present times, medicinal plants occupy the most important position as the source of new drug molecules [4]. Plants and their phytoconstituents have been indispensable in treating diverse forms of diseases, including cancer, bacterial infection, and inflammation [5,6].

Chemopreventive components of plant origin, that are known to target specific cancerous genes or malignant cells, interrupt carcinogenesis and prevent tumor growth, are currently being investigated extensively [7]. For the treatment of over 200 cancer types, only about 140 approved anticancer drugs are available, and there is a lack of effective therapies for most of the tumors [8,9]. Apart from chemotherapeutic drugs, other cancer treatment strategies include radiotherapy, immunotherapy, hormonal therapy, or surgical methods. However, all these strategies are associated with severe side effects, such as damage to normal body cells, disruption of natural metabolic functions, alteration of hormonal balance or immune system etc. and in case of metastatic cancer, chemotherapy is often the only available therapeutic option [8].

Therefore, there is a need to search for alternative and safe drugs and/or methods for the treatment of cancer. Despite the advent of modern drug discovery technologies, such as computer-aided drug design and combinatorial chemistry, isolating novel components from natural products are extremely vital [10]. This is because of the secondary metabolism in plants that naturally evolves in response to the diverse environmental conditions, thus suggesting that a sophisticated natural version of combinatorial chemistry is continually being carried out in plants [11]. The evolution of secondary metabolism in plants has led to the generation of secondary metabolites based on the plants’ requirements, and thus suggest the existence of a remarkable variety and quantity of natural products [12].

Previous studies have demonstrated that cytotoxic phytochemicals either induce apoptosis and necrosis or obstruct variety of cell-signaling pathways, thereby, leading to cell death or cell cycle arrest [13]. Tumor cells possess several molecular mechanisms to alleviate apoptosis and apoptosis suppression, in turn, plays a crucial role in cancer progression. Hence, apoptosis or necrosis induction by cytotoxic agents can be a radical therapeutic approach towards cancer chemotherapy [14].

Considering that over 60% of the antineoplastic drugs originate directly or indirectly (semi-synthetic origin) from plants and most of them are cytotoxic phytochemicals, it is crucial to screen the diverse natural plant resources for new and updated cytotoxic compounds. The Indian subcontinent, known as “the botanical garden of the world”, is of great prominence in this regard as it is one of the 12 regions in the world known to have plants with novel-biomolecules [15,16]. Moreover, approximately 70% of cancer deaths occur in low and middle-income countries like India, Bangladesh, Sri Lanka, Nepal, and Pakistan [8]. Hence, the current investigation was conducted to search for novel cytotoxic plants and active biomolecules in the Indian subcontinent (India, Bangladesh, Sri Lanka, Nepal, and Pakistan) with the hope of supporting new and alternative treatment strategies for cancer chemotherapy.

## 2. Methodology

### 2.1. Search Strategy

An extensive literature search was performed using several platforms such as SciFinder, PubMed, and Google Scholar. The search terms used were “anticancer,” “antitumor,” “cytotoxic,” “plants,” “bioactive compounds,” and “Indian subcontinent”. We focused only on reports in English due to language barrier, time constraints, and translation-related high costs. The obtained bioactive compounds were further evaluated using the database on the Developmental Therapeutic Program (DTP) of the U.S. National Cancer Institute (NCI). In addition, information on in vivo studies as well as reports on pharmacokinetic and clinical trials were included in the review.

### 2.2. Selection of Studies for Inclusion in the Review

The following types of studies were included in this review: (a) in vitro studies on cancer cell lines, (b) in vivo animal studies, (c) studies that indicated the concentrations or doses effective against cancer cells or tumors, and (d) studies that reported the mechanisms of action associated with different compounds. The focus of this review was on potential cytotoxic compounds available in the Indian subcontinent.

### 2.3. Data Extraction

The studies were reviewed with respect to the following information: effects on different cell lines and testing methods, isolated compounds as well as effective compounds, observations, results, suggested mechanisms of action, concentrations tested, and molecular mechanism involved. The chemical structures of the reported bioactive compounds were drawn using the ChemDraw software (version 12.0, PerkinElmer Inc., Waltham, MA, USA).

## 3. Cytotoxic Plants of Indian Subcontinent

In this review, we focused on cytotoxic potential and bioactive compounds obtained from the plants in the Indian subcontinent. A total of 99 cytotoxic plants from 57 families were reviewed and compiled in the current study. The plants were classified according to their families. The bioactive compounds, their respective IC_50_ values (concentration required to inhibit 50% population), and mechanisms of cytotoxicity were listed in Table 1.

### 3.1. Acanthaceae

*Barleria grandiflora*, the only plant belonging to the *Acanthaceae* family with reported cytotoxic potential, demonstrated in vitro and in vivo anti-tumor activity against Dalton’s Lymphoma Ascites (DLA), A-549, and Vero cell lines. Efficacy as an anticancer agent was proved through the prolongation of animal lifespan. An upsurge in ascitic fluid, the major nutritional source of tumor, was witnessed in DLA bearing mice. Treatment with *B. grandiflora* leaves extract in these mice led to the reduction in ascetic fluid as well as cessation of tumor growth, thereby increasing animal lifespan [17,18]. While the most significant adverse effects associated with cancer chemotherapy include anemia and myelosuppression, mice treated with *B. grandiflora* exhibited significant improvement in the red blood cell count and hemoglobin content, and a reduction in white blood cell count, thus demonstrating its antitumor potential [17,19].

### 3.2. Actinidiaceae

*Saurauja roxburghii*, belonging to the Actinidiaceae family, was assessed for cytotoxic activity. Ursolic acid and corosolic acid (Figure 1), obtained from the plant, were tested for cytotoxicity against A431 human epidermoid carcinoma and C_6_ rat glioma cell lines. Corosolic acid was effective against both cancer cells, while ursolic acid was cytotoxic against C_6_ glioma cells [20,21].

### 3.3. Amaryllidaceae

*Narcissus tazetta* was evaluated for cytotoxic potential against MCF-7 (human breast adenocarcinoma) and Hep-2 (human epithelial type 2) cells, and the flower and aerial part extracts led to reduction of cell viability by 40% and 20%, respectively [22].

### 3.4. Anacardiaceae

*Cotinus coggygria* leaf extract was reported to have moderate cytotoxic potential (IC_50_ = 293 µg/mL) against HeLa (Henrietta Lacks cervical cancer) and Vero (Verda Reno) cell lines (IC_50_ >1000 µg/mL) [23]. In addition, the leaf and bark extracts of *Lannea coromandelica* were reported to have cytotoxic effects against AGS (gastric adenocarcinoma), HT-29 (human colorectal adenocarcinoma), MCF-7, and MDA-MB-231 (M. D. Anderson metastatic breast cancer, Houston, TX, USA) cell lines, (IC_50_ values of 90, 520, 270, and 160 µg/mL, respectively) [24]. The cytotoxicity of the plant extract was attributed to the induction of apoptosis and DNA fragmentation and presence of major bioactive phytoconstituents, such as 2-palmitoylglycerol, myricadiol, pyrogallol, and isovanillin. [25,26].

### 3.5. Annonaceae

*Annona muricata* leaf extract was found to be cytotoxic against EACC, MDA, and SKBR3 cancer cell lines (IC_50_ values of 335, 248, and 202 µg/mL, respectively) in in vitro MTT (3-[4,5-dimethylthiazol-2-yl]-2,5-diphenyltetrazolium bromide) assay. Presence of different bioactive alkaloids, flavonoids, terpenoids, and tannins were also reported [27]. The cytotoxic effects of the flower extract of *A. squamosa* were evaluated against MCF-7 and Vero cell lines (IC_50_ values of 6.9 and 75 µg/mL, respectively). The plant was reported to have several bioactive cytotoxic compounds, such as eupafolin, apigenin, and rhamnetin [28].

### 3.6. Apocynaceae

*Tabernaemontana divaricate*, belonging to the Apocynaceae family, was investigated for its cytotoxic potential against NIH3T3 and HeLa cell lines and the respective IC_50_ values were >100 μg/mL and 35.3 μg/mL, respectively. Several alkaloidal cytotoxic compounds i.e., ervachinine, cononitarine, conofoline, conophylline and cisplatin were also reported, among which conophylline (Figure 2) was found to be most potent against HL-60 (human Caucasian promyelocytic leukemia), SMMC-7721, A-549, MCF-7, and SW480 cell lines (IC_50_ values of 0.17, 0.35, 0.21, 1.02, and 1.49 μM, respectively) [29,30].

### 3.7. Araceae

The aerial parts of *Luffa cylindrica* were found to be cytotoxic against MCF-7 and Hep-2 cell lines and the percentage cell viability were 83 and 120, respectively after MTT assay [22]. Another study reported the cytotoxicity of L. *cylindrica* against BT-474, MDA-MB-231, and HepG2 cells, with the IC_50_ value of 329.2 µg/mL against HepG2 [31]. The major cytotoxic compounds were kaempferide, eriodictyol 7-*O*-glucoside, and apigenin 7-*O*-glucouronide (Figure 3), which acted on caspases 3 and 8 [32].

### 3.8. Aristolochiaceae

In the Aristolochiaceae family, *Aristolochia ringens* and *Aristolochia longa* root extracts exhibited cytotoxic effect against multiple cancer cell lines. *A. ringens* was reported to be effective against HeLa, A-431, A-549, PC-3 (prostate cancer), HCT-116, and THP-1 (human acute monocytic leukemia) cells, with IC_50_ values ranging between 3–24 µg/mL, while *A. longa* was found to be active against MCF-7, HT-29, H5-6, and N2A cells, with 22–83% inhibition of tumor cell growth compared to control. Moreover, while testing against S 180 ascites and S 180 solid tumor model, *A. ringens* demonstrated significant tumor growth inhibition in a dose-dependent manner. Furthermore, *A. ringens* also demonstrated a rise in mean survival time (MST) in in vivo models of L1210 lymphoid leukemia [33,34].

### 3.9. Asphodelaceae

Whole plant of *Aloe vera* of Asphodelaceae was examined against HepG2 cell lines and showed time and dose-dependent inhibition of cancer cells (IC_50_ value: 10.5 µg/mL). Cytotoxicity was demonstrated by apoptosis of HepG2 cells through increased expression *p53* and decreased expression of *Bcl-2* genes [35].

### 3.10. Aspleniaceae

A bioactive cytotoxic flavonoid i.e., gliricidin-7-*O*-hexoside was isolated from *Asplenium nidus* (Aspleniaceae family) and evaluated for cytotoxic effect against HepG2 and HeLa cell lines. It demonstrated moderate cytotoxic effect (IC_50_ value of 507 μg/mL) [36].

### 3.11. Asteraceae

Artemisinin (Figure 4), the designated phytoconstituent of *Artemisia annua*, was reported to be active against two human osteosarcoma cell lines, 148B and MG63 (IC_50_ values were 167 μM and 178 μM, respectively) [37]. Dihydroartemisinin (Figure 4), a derivative of the compound was evaluated against canine OSA (Osteosarcoma) cell lines, D-17, OSCA2, OSCA16, and OSCA50 and the respective IC_50_ values were 8.7, 43.6, 16.8 and 14.8 μM [38]. Apart from this, other bioactive compounds of *A. annua* were also tested by evaluating the cytotoxicity of its hydro-alcoholic, dichloromethane, and methanol extracts against D-17, HeLa, and TC221 cell lines, and a dose- dependent anticancer activity was observed [39,40]. For *Bidens pilosa*, different fractions of whole plant were evaluated, using the MTT and Comet assays, against HeLa and KB cell lines. The chloroform fraction was found to be most effective against KB cells, while the water fraction was most active against HeLa cells, with CC_50_ values (concentration of the sample tolerated by 50% of the cultures exposed) of 88.6 and 372.4 µg/mL, respectively [41]. Aerial parts of both *Centaurea antiochia* and *Centaurea nerimaniae* were evaluated against Vero and HeLa cell lines and showed moderate cytotoxicity against *Centaurea antiochia* and significant cytotoxicity against *Centaurea nerimaniae* [23,42].

*Chrysanthemum coronarium* was screened for cytotoxic activity against six cell lines i.e., MCF-7, T47D (ductal epithelial breast tumor), CACO-2 (human colon carcinoma), HRT18, A375.S2, and WM1361A (malignant melanoma), among which WM1361A and T47D were found to be most susceptible (IC_50_ values of 75.7 and 79.8 μg/mL, respectively), while MCF-7 breast carcinoma cell line was least susceptible (IC_50_ = 138.5 μg/mL) [43]. Campesterol (Figure 4), the bioactive steroid isolated from *C. coronarium*, was also reported to induce inhibition of fibroblast growth factor-mediated endothelial cell proliferation [44]. *Inula viscosa* flower extract was evaluated against MCF-7 and Hep-2 cancer cells and demonstrated cytotoxic effect against MCF-7 (IC_50_ value: 15.78 μg/mL) [22].

The leaf extract of *Parthenium hysterophorus* was evaluated for cytotoxicity against DU-145 (prostate), THP-1 (leukemia), MCF-7 (breast), and K562 cell lines, and it was observed that the tumor growth was inhibited at a concentration of about 100 μg/mL [42,45]. The aerial parts of *Phagnalon rupestre* were examined for cytotoxic effects against MCF-7 and Hep-2 cell lines (IC_50_ values: 93.3 μg/mL and 97 μg/mL, respectively) [22]. *Scorzonera tomentosa* extract was tested against Vero and HeLa cell lines (IC_50_ values were 987 and 195 μg/mL, respectively) [23]. *Senecio scandens* was found to be cytotoxic against DL, MCF-7, and HeLa cell lines (IC_50_ values were 27, 24, and 18 μg/mL, respectively) [46].

### 3.12. Berberidaceae

*Berberis aristata* stem extract was found to be cytotoxic against MCF-7 human breast cancer cell (IC_50_ of 220 μg/mL) [47]. The roots of the plant was found to be cytotoxic against DWD (oral), Hop62 (lungs), and A2780 (ovary) cancer cell lines, with the lowest IC_50_ value of 71 μg/mL observed against DWD [48].

### 3.13. Bignoniaceae

*Tecoma stans* was tested against A549 (human lung adenocarcinoma) cell line and cell viability was reduced in a dose-dependent manner, indicating a moderate cytotoxic effect [49]. Another study showed that the plant extract was cytotoxic against HepG2 cells where a reduction in viability of cancer cells was observed [50]. The major anti-proliferative phytoconstituents of the plant included rutin, luteolin, diosmetin, and skytanthine [51].

### 3.14. Boraginaceae

*Cordia dichotoma*, belonging to the *Boraginaceae* family, was tested against prostate carcinoma cell line (PC3). In addition, some bioactive flavonoids were isolated. The cytotoxic activity of the plant extract (IC_50_ value of 74.5 μg/mL) was exerted through apoptosis, nuclear condensation, and ROS (reactive oxygen species) production [52].

### 3.15. Bromeliaceae

*Tillandsia recurvata* was evaluated against A375 (human melanoma), MCF-7, and PC-3 cell lines (IC_50_ values were 1, 40.5 and 6 μg/mL, respectively). Two bioactive dicinnamates i.e., 1,3-di-O-cinnamoylglycerol and (*E*)-3-(cinnamoyloxy)-2-hydroxypropyl 3-(3,4-dimethoxyphenyl) acrylate (Figure 5) were also isolated from the plant. Both of them exhibited activity against A375, PC-3 and MCF-7 cell lines, with IC_50_ values ranging between 3 to 41 μg/mL, respectively [53].

### 3.16. Caesalpiniaceae

Heartwood and leaf extracts of *Caesalpinia sappan* were examined for cytotoxic potential against MCF-7 and A-549 cell lines. Brazilin A (Figure 6) was also isolated and evaluated against MCF-7. Significant cytotoxic property was observed for both crude extracts and bioactive compound. Further molecular docking proved the effectiveness of Brazilin A in the reduction of *Bcl-2* apoptotic inhibitor [54]. *Saraca asoca* (Caesalpiniaceae family) was reported cytotoxic against AGS cell lines (IC_50_ value: 20 μg/mL). Further evaluation of the bioactive compounds was recommended [24].

### 3.17. Compositae

*Gnaphalium luteoalbum*, belonging to the *Compositae* family, was evaluated for cytotoxic effects against VERO, NIH3T3, AGS, HT-29, MCF-7, and MDA-MB-231 cell lines and was found to be cytotoxic against AGS and MCF-7 cells (IC_50_ values of 980 and 340 μg/mL). The major cytotoxic phytoconstituents included apigenin, luteolin, jaceosidin, and gnaphalin [24,55].

### 3.18. Dilleniaceae

The stem and bark extracts of *Dillenia pentagyna* (Dillenaceae family) were found to have cytotoxic activity against DL, MCF-7, and HeLa cell lines (IC_50_ values of 25.8, 41.6 and 76.8 µg/mL, respectively). Reduction in glutathione (GSH) level was also observed in *D. pentagyna* treated animal. GSH is related with onset of tumor cell proliferation through regulation of PKC (protein kinase C). Thus, *D. pentagyna* was reported to be beneficial in cancer treatment through depletion of GSH [46,56]. In addition, *Dillenia indica* leaves have also been reported to be cytotoxic against MCF-7 and MDA-MB-231 cell lines (IC_50_ of 340 and 540 μg/mL, respectively) [24].

### 3.19. Dipsacaceae

The aerial parts of *Pterocephalus pulverulentus* demonstrated moderate cytotoxic effects against MCF-7 and Hep-2 cell lines (percentage of remaining viable cells was 80 and 138%, respectively) [22].

### 3.20. Ebenaceae

*Diospyros peregrina* leaves were reported to have cytotoxic activity against MCF-7 and MDA-MB-231 cell lines (IC_50_ values were 7 and 33 μg/mL, respectively) [24]. Another study reported the cytotoxicity of different fractions of *D. peregina* fruits against MCF-7 and HepG2 cell lines (lowest IC_50_ values were 37.2 and 64.4 μg/mL, respectively) [57].

### 3.21. Ericaceae

*Arbutus andrachne* aerial part was reported to have cytotoxic activity against MCF-7, T47D, CACO-2, HRT-18, A375.S2, and WM1361A cell lines. Significant cytotoxic effect was observed against A375.S2, HRT-18, and MCF-7 cell lines, while moderate cytotoxic potential was observed for other cell lines. The percentage of remaining cell viability was within 40.56 to 121.2% [43].

### 3.22. Euphorbiaceae

*Croton caudatus* was evaluated against DL, MCF-7, and HeLa cell lines. Potent cytotoxic effect against DL cell line was observed (IC_50_ value: 29.7 µg/mL) [46]. The cytotoxic effect of *Euphorbia tirucalli* was evaluated against pancreatic cancer cell line (MiaPaCa-2) and a dose-dependent cytotoxic effect was observed with a reduction in quantity of viable cells (7% for 200 µg/mL) [58].

### 3.23. Fabaceae

*Adenanthera pavonina* of family Fabaceae was evaluated in vivo for its cytotoxic potential in DLA induced ascetic mice model. Significant reduction in volume of tumor and number of viable tumor cells were observed in mice treated with the plant extract [18]. For *Ononis sicula* and *Ononis hirta*, the antiproliferative activity of different fractions of aerial parts were evaluated against MCF-7, Hep-2 and Vero cell lines. For *Ononis hirta*, the lowest IC_50_ values were 28 (methanol fraction), 48.8 (chloroform fraction), and 41.9 μg/mL (n-hexane fraction) against MCF-7, Hep-2 and Vero cell lines, respectively. On the other hand, the chloroform fraction of *Ononis sicula* was most effective against MCF-7, Hep-2, and Vero cell lines (IC_50_ values of 66, 75.3, and 79.5 μg/mL, respectively) [22].

### 3.24. Gramineae

Leaves of *Zea mays* (*Germineae* family) were reported to be cytotoxic against Hep-2 cell lines and resulted in an increase in the number of apoptotic Hep-2 cells [59]. Maysin, a cytotoxic constituent isolated from the plant, demonstrated moderate cytotoxic effects against five cancer cells namely, A549, SK-OV-3 (ovarian), SK-MEL-2 (melanoma), XF-489 (CNS), and HCT-15 (colon) (IC_50_ of 62.24, 43.18, 16.83, 37.22, and 32.09 μg/mL, respectively) [60].

### 3.25. Hypericaceae

*Hypericum kotschyanum* aerial part was reported to have moderate cytotoxic effects against HeLa and Vero cell lines (IC_50_ values were 507 and 367 μg/mL, respectively) [23].

### 3.26. Labiatae

The aerial parts of *Salvia pinardi* were significantly cytotoxic against MCF-7 and Hep-2 cell lines (IC_50_ of 85.5 (methanol fraction), 94.8 (chloroform fraction), and 192.3 (n-hexane fraction) μg/mL) [22].

### 3.27. Lamiaceae

*Lavandula angustifolia*, belonging to the *Lamiaceae* family, was found to have negligible or no activity against MCF-7 and Hep-2 cell lines [22]. On the other hand, *Plectranthus stocksii* stem and leaf extract was significantly cytotoxic against RAW264.7, Caco-2, and MCF-7 cell lines. The least IC_50_ value for RAW264.7 was 9 mg/mL (stem ethyl acetate fraction), for Caco-2 it was 36.1 mg/mL (leaves ethyl acetate fraction), and for MCF-7, it was 48.9 mg/mL (leaf ethyl acetate fraction) [61]. For *Salvia hypargeia*, cytotoxic activity was examined against Vero and HeLa cell lines and the IC_50_ values were over 1000 µg/mL against both cell lines [23]. In contrast, *Salvia officinalis* was found to be significantly cytotoxic against MCF-7, B16F10, and HeLa cells (IC_50_ = 14–36 µg/mL). Two bioactive compound i.e., α-humulene and trans-caryophyllene (Figure 7) were also isolated and tested against MCF-7 cell line and the IC_50_ values were found to be 81 and 114 μg/mL, respectively [62]. the aerial parts of *Teucrium sandrasicum* were examined for cytotoxic effects against HeLa and Vero cell lines and the reported IC_50_ values were 513 and 593 µg/mL, respectively [23].

*Nepeta italica*, belonging to Lamiaceae family, was found to have moderate cytotoxicity against HeLa and Vero cell lines (IC_50_ values of 980 and >1000 μg/mL, respectively) [23]. In a DMBA (7,12-dimethylbenz[a]anthracene)-induced hamster buccal pouch carcinogenesis model, *Ocimum sanctum* demonstrated inhibition of tumor development and early events of carcinogenesis [63]. In S 180 induced mice model, an increase in survival rate was observed without any impact on tumor volume. These effects were attributed to its indirect or direct impact on the immune system through modulation or regulation of humoral immunity and stimulation of cell-mediated immunity, thereby resulting in inhibition of neoplasm [64,65,66]. *Origanum sipyleum* was found to have minimal cytotoxic effect against both HeLa and Vero cell lines, with IC_50_ values over 1000 μg/mL. [23]. *Teucrium polium* (*Laminaceae* family) was examined against MCF-7, Hep-2, T47D, CACO-2, HRT18, A375.S2, and WM1361A cell lines. The extract was found to be most effective against MCF-7, Hep-2, and A375.S2 cell lines (percent of remaining viable cells were 78, 58, and 61%, respectively) [22,43].

### 3.28. Malvaceae

Three species of *Hibiscus* (Malvaceae family) i.e., *Hibiscus micranthus*, *Hibiscus calyphyllus,* and *Hibiscus deflersii* were evaluated for their cytotoxic effect against HepG2 and MCF-7 cell lines, using the MTT assay. The petroleum ether fraction of *H. deflersii* extract was found to be most potent amongst the three species against HepG2 and MCF-7 cell lines (IC_50_ values were 14.4 and 11.1 μg/mL, respectively). Three cytotoxic biomarkers i.e., ursolic acid, β-sitosterol, and lupeol (Figure 8) were quantified through HPLC analysis and highest concentration of these biomarkers were obtained from the petroleum ether fraction of *H. deflersii* extract.

Again, the petroleum ether fraction of *H. calyphyllus* extract (IC_50_ values were 14.5 and 25.1 μg/mL against HepG2 and MCF-7, respectively) and the chloroform fraction of *H. micranthus* extract (IC_50_ values were 27.6 and 24.1 μg/mL against HepG2 and MCF-7, respectively) had the most potent cytotoxic effect [67]. Both extracts were then analyzed with HPLC for the quantification of the aforementioned biomarkers. Among the three biomarkers, the apoptotic effect of ursolic acid was reported to be mediated by cytochrome c-dependent caspase-3 activation, inhibition of DNA replication through topoisomerase I cleavage, and increase in the expression of p21^WAF1^ cell-cycle regulator [68]. β-Sitosterol induced apoptosis via caspase-3, caspase-9 activation and poly (ADP-ribose)-polymerase cleavage. In addition, reduction in anti-apoptotic *Bcl-2* protein expression and increase in pro-apoptotic *Bax* protein expression were reported [69]. The cytotoxic effects of lupeol were attributed to the inhibition of topoisomerase II, DNA polymerase, angiogenesis, and induction of apoptosis through caspases activation, poly (ADP-ribose)-polymerase cleavage, and decreased *Bcl-2* expression [70].

### 3.29. Meliaceae

*Azadirachta indica*, an important medicinal plant found on the Indian subcontinent, was evaluated for its cytotoxic effect against HT-29, A-549, MCF-7, HepG-2, MDBK, and EAC cell lines (IC_50_ values: 83.5–212.2 μg/mL) [71,72]. Multiple cytotoxic triterpenoids, limonoids (tetranortriterpenoids), and flavonoids were isolated from *A. indica*, and their cytotoxic effects have been reported in numerous studies. One study reported the isolation of 17 limonoids (including three new compounds) and assessed them for cytotoxic potential against lung (A549), leukemia (HL60), breast (SK-BR-3), and stomach (AZ521) cancer cell lines. Seven compounds exhibited significant cytotoxic potential (IC_50_ values: 0.1–9.9 μM). Among them, nimonol (Figure 9) induced apoptosis in HL60 cells (IC_50_ value: 2.8 μM) [73]. In another study, 36 limonoids were isolated (including six new compounds) and the anti-tumor potential was evaluated against the aforementioned cell lines (IC_50_ 0.1–9.3 μM). Sixteen compounds were reported to induce apoptosis in AZ521 cells [74,75]. In addition, two new flavonoids, 3′-(3-hydroxy-3-methylbutyl) naringenin and 4′-O-methyllespedezaflavanone C (Figure 9), and seven known flavonoids were isolated from the plant and were found to be potentially cytotoxic against A549, HL60, SK-BR-3, and AZ521 cell lines (IC_50_ 4.2–100 μM) [76]. Furthermore, two new terpeniods, isolated from *A. indica*, were found to be cytotoxic against A549, HL60, SW-480, SMMC7721, and MCF-7 cells (IC_50_ values: 0.8 to 4.5 μM) [77].

### 3.30. Menispermaceae

The aerial parts of *Cocculus hirsutus* were evaluated for cytotoxic effects against MCF-7 cell line using in vitro MTT assay. The methanol extract of the plant demonstrated cytotoxic potential (IC_50_ value: 39.1 μg/mL). Multiple bioactive anticancer compounds were isolated from this plant, including coclaurine, haiderine, and lirioresinol, which exerted anti-tumor effects by interacting with cell-cycle regulatory proteins, such as Aurora kinase, c-Kit, FGF, Nuclear Factor-Kappa B (*NF-kB*), *Bcl-xL*, and VEGF [78,79].

### 3.31. Moraceae

*Artocarpus heterophyllus*, belonging to the *Moraceae* family, was examined for cytotoxic effect against HEK293, A549, HeLa, and MCF-7 cell lines using the MTT and sulforhodamine B (SRB) assays. The plant extract was found to have cytotoxic effect against A549 cell line (IC_50_ value of 35.3 μg/mL) [80]. In other studies, nine bioactive flavonoids were isolated and evaluated against B16 melanoma cells and T47D cells. The compounds with significant cytotoxic effects were noratocarpin, cudraflavone, artocarpin, brosimone, kuwanon, and albanin. (Figure 10), with IC_50_ values ranging between 7 to 32 μg/mL [81,82]. The effect was reported to be due to inhibition of melanin biosynthesis [83]. Among the three *Ficus* species, i.e., *F. beecheyana*, *F. carica,* and *F. racemosa*, *F. beecheyana* led to a dose-dependent reduction in cell viability of AGS, SW-872, HL-60, and HepG2 cells due to cellular apoptosis. Apoptosis was thought to be induced by polyphenolic compounds, such as p-coumaric acid, chlorogenic acid, caffeic acid, gallic acid, and rutin through interaction with Fas, Fas-L, *p53*, *Bcl-2*, and caspases (3,8,9) proteins [84]. In addition, *F. carica* was found to have cytotoxic activity against MCF-7, B16F10, and HeLa cell lines (IC_50_ values were 440, 880, and >1000 μg/mL, respectively) [62].

Different parts of *F. racemosa* extracts were evaluated for cytotoxic effects against MCF-7, DLA, HL-60, HepG2, NCI-H23, and HEK-293T cell lines. Cytotoxicity was observed against MCF-7, DLA, HL-60, and HepG2 cells (IC_50_ values were 80, 175, 276.9, and 363 μg/mL, respectively) [85,86,87]. The extract of *Morus nigra* was evaluated for cytotoxic effects against OVCAR-8 (ovarian), SF-295 (brain), HCT-116 (colon), and HeLa (cervix) cancer cells, and demonstrated significant cytotoxicity against HeLa (% inhibition 89.5–32) [88,89].

### 3.32. Myristicaceae

*Myristica fragrans* was found to have cytotoxic activity against KB cell lines (IC_50_ value: 75 μg/mL). Cytotoxicity was induced by apoptosis of cancer cells through interaction with *Bcl-2* protein [10]. Other studies demonstrated 2 new phenolic and 38 essential oils from *M. fragrans* which were tested against K-562, HCT-116, and MCF-7 cells, and the IC_50_ values were found to be within the range 2.11–78.15 μg/mL [90,91].

### 3.33. Myrtaceae

*Syzygium cumini*, belonging to the Myrtaceae family, was reported to have cytotoxic effects against A2780, MCF-7, PC3, and H460 cell lines (IC_50_ values were 49, 110, 140, and 165 μg/mL, respectively) [92]. In addition, the cytotoxic effects of *S. cumini* against U251, HepG2, and HeLa cell lines were also reported [93,94]. The major bioactive components in *S. cumini* were anthocyanins (pelargonidin-3-*O*-glucoside, pelargonidin-3,5-*O*-diglucoside, cyanidin-3-*O*-malonyl glucoside, and delphenidin-3-*O*-glucoside), tannins (ellagic acid, ellagitannins etc.), and polyphenols [92,93].

### 3.34. Nyctaginaceae

The aerial parts of *Mirabilis jalapa* were tested for cytotoxic activity against MCF-7 and Hep-2 cell lines and the percentage of remaining viable cells after treatment were 60 and 78, respectively [22]. Studies have reported the presence of bioactive rotenoids and Mirabilis antiviral protein (MAP) in *M. jalapa*, which were found to be cytotoxic against HeLa, Raji, A549, HCT 116, and Vero cell lines. The IC_50_ values of MAP against HCT116, MCF-7, and A549 cell lines were 150, 175, and 200 µg/mL, respectively [95,96].

### 3.35. Oleaceae

The flowers of *Jasminum sambac* were observed to have moderate cytotoxicity against MCF-7 and HEP-2 cell lines [22]. Other studies reported the cytotoxicity of the leaves of *J. sambac* against MCF-7 cell lines (IC_50_: 7 µg/mL) [24]. *Olea europaea* leaves have also been reported to have cytotoxic activity against MCF-7, B16F10, HeLa, and Vero cell lines (IC_50_ values were 43, 170, 440, and >1000 µg/mL, respectively) [23,62]. Cytotoxicity of *O. europaea* was due to the induction of apoptosis through interaction with *Bcl-2*, *Bax*, and *p53* proteins [97]. The fruits of *Syringa vulgaris* were found to be cytotoxic against MCF-7 and HeLa cell lines (viability of cells were 71 and 122%, respectively) [22].

### 3.36. Oxalidaceae

The leaves and fruits of *Averrhoa bilimbi* were found to be cytotoxic against MCF-7 breast cancer cell lines (IC_50_ was 154.9 and 668 µg/mL for fruits and leaves, respectively) [98]. The major bioactive phytoconstituents included nonanal, tricosane, squalene, and malonic acid [99].

### 3.37. Phyllanthacae

*Phyllanthus emblica*, belonging to the *Phyllanthacae* family, was found to be cytotoxic against HT-29 colon cancer cell lines (IC_50_ value: ~35 µg/mL) [100]. Amongst the bioactive compounds, a new apigenin glucoside and 14 sterols have been isolated (including two new compounds) and screened for cytotoxic activity against HL-60 and SMMC-7721. Among these compounds, trihydroxysitosterol (Figure 11) exhibited cytotoxicity against both HL-60 and SMMC-7721 (IC_50_ values of 85.56 and 165.82 µg/mL, respectively) [101,102].

### 3.38. Plumbaginaceae

*Limonium densiflorum* halophyte was reported to have cytotoxicity against A-549 and DLD-1 cancer cell lines (IC_50_ values were 29 and 85 μg/mL, respectively). The major bioactive phytoconstituents included myricetin, isorhamnetin, and *trans* 3-hydroxycinnamic acid [103].

### 3.39. Picrorhiza

*Picrorhiza kurroa* rhizomes, belonging to the *Picrorhiza* family, were found to have cytotoxic effects against Hep3B (hepatocellular carcinoma), MDA-MB-435S (breast carcinoma), and PC-3 (prostate) cancer cell lines (lowest IC_50_ values were 32.6, 19, and 63.9 μg/mL, respectively). Cytotoxicity was induced in the target cells through apoptosis [104]. Among the phytoconstituents isolated from the plant, curcubitacin was a potent cytotoxic and antitumor agent [105].

### 3.40. Pinaceae

The cytotoxic activity of *Cedrus deodara*, belonging to the *Pinaceae* family, was tested against multiple cancer cell lines. It was found to be cytotoxic against Mia-Pa-Ca-2, PC3, and A-2780 (ovary) cancer cells (IC_50_ of 74, 77, and 63 μg/mL, respectively) [48]. In addition, the plant extract was found to have cytotoxic potential against Molt-4 cancer cells (IC_50_ value: 15 μg/mL). Cytotoxicity was induced by apoptosis of cancer cells through interaction with caspase 3, 8, and 9 proteins [106].

### 3.41. Piperaceae

The cytotoxicity of *Piper longum*, belonging to the *Piperaceae* family, was evaluated against multiple cancer cell lines. Significant cytotoxic effects were observed against Colo-205 colon cancer cell lines (IC_50_ was 18 μg/mL) [48]. Other studies reported cytotoxicity of *P. longum* fruits against DLA and EAC cancer cells. The major cytotoxic phytoconstituent was piperine and its antitumor effect was due to the induction of apoptosis [107]. The cytotoxicity of *P. regnelli* was tested against melanoma (UACC-62), kidney (786-0), breast (MCF-7), lung (NCI-H460), ovary (OVCAR-3), prostate (PC-3), colon (HT-29), and leukemia (K-562) cancer cells. The major bioactive agent of *P. regnelli* was eupomatenoid-5 and it demonstrated cytotoxicity against PC3, OVCAR-3, 786-0, and MCF-7 cells [108].

### 3.42. Poaceae

The aerial parts and roots of *Cenchrus ciliaris* were evaluated for cytotoxic activity against A-549, CACO, HCT-116, HeLa, HepG2, MCF-7, and PC3 cell lines. Significant cytotoxicity was observed against HepG2 (IC_50_ values were 12 and 9 µg/mL for aerial parts and roots, respectively), CACO (IC_50_ values were 27.2 and 20.5 µg/mL for aerial parts and roots, respectively), and A-549 (IC_50_ values were 14.5 and 11.1 µg/mL for aerial parts and roots, respectively) [109,110].

### 3.43. Polygonaceae

*Calligonum comosum*, belonging to the *Polygonaceae* family, was found to be cytotoxic against HepG2 cancer cell lines (IC_50_ value: 9.60 µg/mL). The major cytotoxic phytoconstituents of the plant were catechin and its derivatives as well as kaempferol and its derivatives, including mequilianin. Cytotoxicity was induced by apoptosis through increased expression of *p53* and reduced expression of *Bcl-2* gene [35,111].

### 3.44. Primulaceae

*Aegiceras corniculatum*, a well-known medicinal plant belonging to the *Primulaceae* family, was found to have significant cytotoxic effect against VERO, NIH3T3, AGS HT-29, MCF-7, and MDA-MB-231 cell lines. (IC_50_ values of 150, 97, 0.5, 998, 91, and 461 μg/mL, respectively) [24]. The isolated bioactive constituents were aegicoroside A, sakurasosaponin, fusarine, and fusamine. [112,113].

### 3.45. Ranunculaceae

The seeds of *Delphinium staphisagria* (*Ranunculaceae* family) were found to be cytotoxic against MCF7, HT29, N2A, H5-6, and VCREMS cell lines. In addition, cytotoxicity was observed against N2A, H5-6, and VCREMS (IC_50_ values were 41.9, 41.1, and 14.5 μg/mL, respectively) [34]. The major constituents of the plant were astragalin, paeonoside, and petiolaroside. [114].

### 3.46. Rosaceae

The leaves of *Crataegus microphylla*, belonging to the *Rosaceae* family, were found to have cytotoxic activity against HeLa and Vero cell lines (IC_50_ of 576 and >1000 μg/mL, respectively) [23]. Another study reported cytotoxicity of the flowers of *C. microphylla* against HeLa cells (IC_50_ 871 μg/mL) [115]. The cytotoxic effects of *Rosa damascene* were evaluated against MCF-7, Hep-2, HeLa, and Vero cell lines, and significant cytotoxic activity was observed against HeLa cells (IC_50_ value of 265 μg/mL) [22,23]. The cytotoxic constituents of *R. damascena* included nerol, geraniol, β-citronellol, linalool, nonadecane, and phenylethyl alcohol [116].

### 3.47. Rubiaceae

The bark and wood of *Hymenodictyon excelsum*, belonging to the *Rubiaceae* family, were screened for cytotoxicity against Vero, NIH3T3, AGS HT-29, MCF-7, and MDA-MB-231 cell lines, and significant cytotoxic effects of the bark were observed against all cell types (IC_50_ values were 230, 70, 90, 160, 80, and 440 μg/mL, respectively) [24]. The cytotoxicity was induced by DNA fragmentation and apoptosis of cancer cells [117]. The leaves of *Oldenlandia corymbosa* were reported to be cytotoxic against K562 cells (IC_50_ of 114.4 μg/mL) and the toxicity was induced by apoptosis [118].

### 3.48. Salicaceae

The flowers of *Populus alba* were reported to be moderately cytotoxic against MCF-7 and Hep-2 cells. The percentage of residual cell viability was 100% for both cell lines at 100 μg/mL dose [22]. Another study revealed the cytotoxicity of the essential oils from the plant against A549, H1299 (human non-small cell lung cancer), and MCF-7 cancer cells (IC_50_ values were 12.05, 10.53, and 28.16 μg/mL, respectively) [119].

### 3.49. Sapotaceae

Flowers of *Manilkara zapota*, belonging to the *Sapotaceae* family, were found to be cytotoxic against MCF-7 cell line (IC_50_ was 12.5 μg/mL) via DNA fragmentation [28]. Eleven triterpenoid saponins, including lupanes, oleananes, ursanes, and manilkoraside were isolated from the plant, among which manilkoraside exhibited the highest cytotoxic effects against HT-29 and HL-60 cell lines (EC_50_ of 64 and 24 μg/mL, respectively) [120].

### 3.50. Saururaceae

*Saururus chinensis* roots were found to be cytotoxic against MCF-7 breast cancer cell lines in the MTT assay. The lowest IC_50_ value calculated was 91.2 μg/mL (water fraction). The major bioactive constituents included aristolactram, dihydroguaiatric acid, and sauchinone [121]. Saucerneol D, manassantin A and B, and saucerneol F were some of the lignans isolated from *S. chinensis*, which had cytotoxic effects against HT-29 and HepG2 cells (IC_50_: 10–16 μg/mL). The cytotoxicity was due to the inhibition of DNA topoisomerase I and II [122].

### 3.51. Scrophulariaceae

The flowers and aerial parts of *Verbascum sinaiticum* were reported to have moderate cytotoxicity against MCF-7 and Hep-2 cancer cells (cell viability of 60 and 80%, respectively [22]. Another study reported cytotoxic effects of two flavonolignans, novel sinaiticin and hydrocarpin, obtained from *V. sinaticum*. They were tested against P-388 cells and found cytotoxic with ED_50_ of 1.2 and 7.7 μg/mL for hydrocarpin and sinaiticin, respectively [123].

### 3.52. Solanaceae

*Solanum khasianum*, belonging to the *Solanaceae* family, was screened for cytotoxicity against DL, MCF-7, and HeLa cancer cells (IC_50_ values were 27.4, 71.2, and 62.5 μg/mL, respectively) [46]. The cytotoxic effects of the fruits of *S. nigrum* were evaluated against HeLa, Vero, HepG2, and CT26 cancer cell lines (IC_50_ values were 265, 6.9, 56.4, and 77.6 μg/mL, respectively) [124,125].

Two species of *Withania* i.e., *W. coagulans* and *W. somnifera* were also evaluated for their cytotoxic potential. The fruits, roots, and leaves of *W. coagulans* were screened for cytotoxic effects against HeLa, MCF-7, and RD cancer cell lines. The IC_50_ values for leaves and roots were 0.7–4.7 μg/mL and 0.7–6.7 μg/mL, respectively. The major constituents were myricetin, quercetin, and gallic acid [126]. *W. somnifera* demonstrated cytotoxicity against PC3, A-549, A-2780, and K-562 cell lines (IC_50_ of 52, 46, 79, 41 μg/mL, respectively) [48].

### 3.53. Sterculiaceae

*Helicteres isora* whole plant was screened for cytotoxicity against HeLa-B75, HL-60, HEP-3B, and PN-15 cells. Moderate effectiveness was observed against all cell types. The major cytotoxic components of this plant were cucurbitacin B and isocucurbitacin B [127,128].

### 3.54. Thymelaeaceae

*Aquilaria malaccensis*, belonging to the *Thymelaeaceae* family, was found to be cytotoxic against DLA and EAC cancer cells (IC_50_ values were 72 and 79 μg/mL, respectively). However, the cytotoxicity against normal cells was negligible [129]. Another study evaluated the cytotoxicity of the oil fraction of the plant extract against HCT116 colon cancer cells and reported an IC_50_ value 4 μg/mL [130]. The major phytoconstituents of the plant were benzaldehyde, pinene, octanol, germacrene, and hexadecanal [131].

### 3.55. Verbenaceae

*Clerodendrum viscosum* leaf extract was screened against VERO, NIH3T3, AGS, HT-29, MCF-7, and MDA-MB-231 cancer cell lines and was found to have cytotoxic effects against MCF-7 and HT-29 cells (IC_50_ of 50 and 880 μg/mL, respectively) [24]. The roots of *C. infortunatum*, belonging to the same family, exhibited in vivo cytotoxicity in DLA-induced ascetic mice model and the induction of apoptosis was observed through interaction with *Bax*, *Bcl-2*, caspases 8, and 10 proteins [132].

### 3.56. Vitaceae

*Leea indica*, belonging to the *Vitaceae* family, was found to be cytotoxic against DU-145 and PC-3 prostate cancer cell lines (IC_50_ values were 529.4 and 677.1 μg/mL, respectively) [133]. In case of *Vitis vinifera*, the stem was screened against MCF-7, B16F10, and HeLa cancer cell lines and the lowest IC_50_ values were 62, 137, and 336 μg/mL, respectively for the acetone fraction of the extract [62]. Two major cytotoxic polyphenols in *V. vinifera* i.e. quercetin 3-*O*-β-d-4C_1_ galactoside and quercetin 3-*O*-β-d-4C_1_ glucuronide, were isolated. The cytotoxic effects were induced through apoptosis [134,135].

### 3.57. Zingiberaceae

Rhizomes of *Curcuma longa*, belonging to the *Zingiberaceae* family, were found to have 97% cytotoxicity against Hep-2 cell line, at a dose of 1000 μg/mL [136]. The major phytoconstituent of this plant is curcumin (Figure 12), which has been reported to have anticancer activity in previous studies [137,138]. Amongst other phytoconstituent in *C. longa*, β-sesquiphellandrene (Figure 13) has also been reported to possess anticancer potential [139]. The cytotoxic effect of curcumin was through the induction of apoptosis [140].

The major constituents of *Zingiber officinale*, i.e., gingerol (Figure 13) and its derivatives were screened against A549, SK-OV-3, SK-MEL-2, and HCT15 and were found to be cytotoxic (IC_50_ < 50 μM). Many essential oils of *Z. officinale* have also been reported to have cytotoxic effects [141,142]. The combination effect of *C. longa* and *Z. officinale* were observed to have synergistic cytotoxic effect against PC-3M cancer cells [143].

## 4. Reported Cytotoxic Constituents: Therapeutic Perspective and Future Directions

The aforementioned cytotoxic phytoconstituents, isolated from different plants of Indian subcontinent, were further examined for their significance in cancer chemotherapy. The DTP database of the National Cancer Institute (NCI), USA, was utilized to assess the therapeutic effects of these bioactive constituents [144].

Ursolic acid, a pentacyclic triterpenoid, was found to have significant anticancer effect against MCF-7 and MDA-MB-231 breast cancer cell lines. The reported mechanisms of cytotoxicity included reduction in cyclin D1, STAT3 (signal transducer and activator of transcription), CDK4 (cyclin dependent kinase), *Bcl-2*, AKT, and MMP-2 (matrix metallopeptidases) proteins as well as activation of *Bax*, caspase 3, caspase 8, caspase 9, PARP (poly(ADP-ribose)polymerase), *p53*, and *p21* proteins (Figure 13 and Figure 14) [145]. DTP database revealed several in vivo animal model studies against L1210, P388, and B16 cancer cells. Although the compound showed low toxicity in in vivo studies, the major impediment in the therapeutic development of ursolic acid was its poor bioavailability and short plasma-half life. Currently, attempts are being made to improve its pharmacokinetic parameters by utilizing nano-particle-based drug delivery techniques [146].

Artemisinin, a well-known anti-malarial drug and its derivative dihydroartemisinin, were found to have anticancer activity in numerous in vitro and in vivo studies. The cytotoxic effects were attributed to DNA damage by base excision or homologous recombination, cell death by apoptosis, autophagy, and necrosis; inhibition of angiogenesis via AMPK (AMP activated protein kinase) pathway; reduction in CyclinD1, CyclinE, and *Bcl-2*; and activation of *Bax*, Caspase 3, Caspase 8, Caspase 9, PARP, *p53*, and *p21* (Figure 13 and Figure 14). Currently, clinical trials are being conducted to establish artemisinin as a potential anticancer agent [147].

Brazilin was reported to have cytotoxic effects against TCA8113, MG-63, and T24 cancer cells. The reported mechanisms of cytotoxicity included interaction with c-Fos, inhibition of *Bcl-2*, *p62*, and p-mTOR as well as enhancement of *Bax*, caspase-3, LC3B, and p-AMPK (Figure 13 and Figure 14) [148,149]. Future studies should focus on the pharmacokinetic evaluation of this compound.

β-Sitosterol was found to be have anti-cancer effects against breast, prostate, lung, colon, stomach, and ovarian cancers as well as leukemia. The established mechanisms of cytotoxicity were through interaction with cell signaling pathway, apoptosis, invasion, angiogenesis, metastasis, and proliferation. Although the compound was reported to be nontoxic, it was found to be less potent. The use of cell-specific or liposome-based drug delivery is proposed for the successful application of this compound as an anticancer agent [150].

Lupeol was reported to be cytotoxic against HeLa, CWR22Rt1, A549, 451Lu, SMMC7721, and WM35 cancer cells by downregulation of cyclin D1, CDK2, and *Bcl-2* and upregulation of *Bax*, caspase-3, and *p38* (Figure 13 and Figure 14). The compound is currently under investigation in various clinical trials, and if successful, it might be used as a novel adjuvant therapeutic agent for treating multiple cancers in human [151].

Curcumin is a prominent natural phytoconstituent used in the treatment of several types of cancers, including prostate, pancreatic, colorectal, breast, and lung cancers as well as multiple myeloma and leukemia. Reported molecular targets for curcumin include *Bcl-2*, *Bax*, *p53*, caspases, *p38*, *NF-kB*, PARP, and cyclin proteins (Figure 13 and Figure 14). The limited bioavailability of curcumin, owing to poor absorption and rapid metabolism, has been improved by synthesizing structural analogues and development of liposomal, nanoparticle, and phospholipid complex-based drug delivery systems. Several clinical trials have been conducted to evaluate the efficacy of curcumin. Further studies are required to enhance the bioavailability of this compound [152].

A large number of studies have reported the cytotoxicity of catechins, such as epicatechin, epigallecatechin-3-gallate etc. Possible mechanisms of cytotoxicity include interaction with *Bcl-2*, STAT, FYN, *p53*, CDKs, and vascular endothelial growth factor (VEGF). Several in vivo studies have also demonstrated the effectiveness of catechins in animal models. Additional pharmacokinetic and pharmacodynamic studies in humans are required [153].

Myricetin is a natural flavonoid that has extensively been reported for its antitumor and cytotoxic potential against gastric, esophageal, ovarian, colon, and cervical cancers and multiple leukemia, melanoma, and sarcoma. The compound was tested against many cell lines, including HGC-27, MCF-7, HCT-15, OVCAR-3, HepG2, A549, HeLa, and PC3. Myricetin was found to interact with multiple proteins, such as *Bcl-2*, *Bax*, Caspases, *Bcl-xl*, *p53*, *p21*, *NF-kB*, and cyclin, IL (Figure 13 and Figure 14) [154]. Despite being a potential cytotoxic agent, poor aqueous solubility and poor absorption of this compound has led to its reduced bioavailability. Formulation into nano-suspensions or micro-emulsions was proposed to enhance its absorption. More studies are required in this regard [155].

Quercetin, another naturally occurring polyphenolic flavonoid, has already been reported to be effective against breast, colon, pancreatic, lung, liver, prostate, bladder, gastric, bone, blood, brain, cervical, skin, eye, ovarian, thyroid, and kidney cancers. The cytotoxicity of the compound has been attributed to its interaction with *p51*, *p21*, caspases, TNF, IL, *Bcl-2*, *Bax*, *p53*, myeloid cell leukemia (MCL), and cyclin proteins (Figure 13 and Figure 14) [156]. Despite of its low toxicity, multiple clinical trials have been conducted with quercetin. However, extensive studies are required before the development of an established anticancer quercetin formulation [157].

Among the other reported phytoconstituents, caryophyllene, campesterol, rutin, gallic acid, and caffeic acid have been assessed in vivo by DTP of NCI, USA in mice models of P388, L1210, Friend virus leukemia, and Lewis lung carcinoma. However, data regarding the remaining bioactive cytotoxic phytoconstituents reported in this review article, is very limited. Hence, in future, further in vivo as well as mechanistic studies are warranted to assess the activity of these cytotoxic compounds.

Moreover, out of almost 50,000 plant species found in the Indian subcontinent, only a handful of them have been properly examined for their cytotoxic potentials [158]. The diverse environmental conditions such as dry forests, mangrove forests, deserts, aquatic reservoirs, and hill tracts have made the Indian subcontinent a natural treasure. However, the studies conducted so far have just explored the tip of the iceberg. Therefore, the Indian subcontinent still remains a vast unexplored region with great number of exclusive medicinal plants. Further studies should investigate these endemic plants with respect to their cytotoxic potential, mechanism of action, and isolation of active constituents.

## Figures and Tables

**Figure 1 molecules-25-01904-f001:**
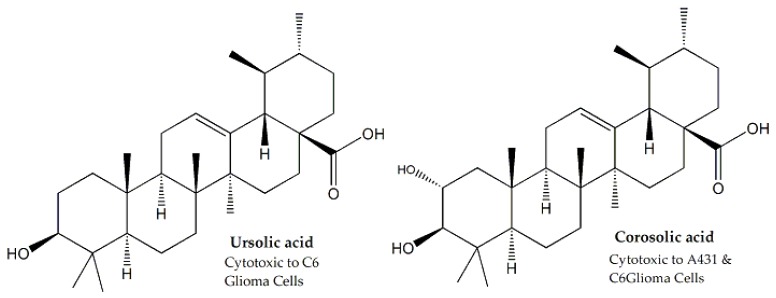
Chemical structure and cytotoxic potential of ursolic and corosilic acid.

**Figure 2 molecules-25-01904-f002:**
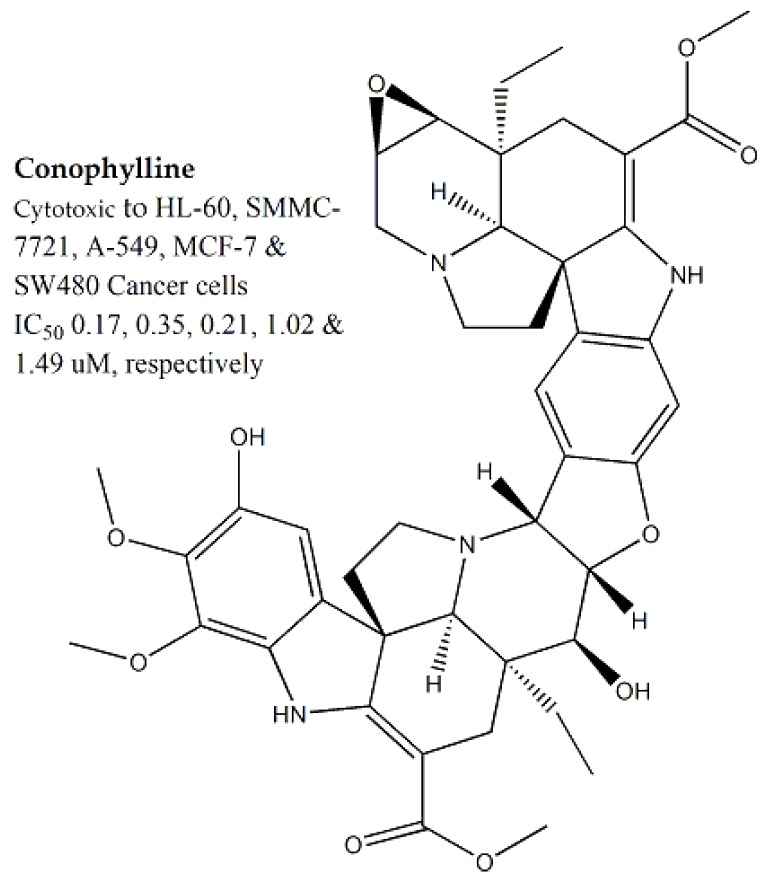
Cytotoxic bioactive constituent from *Tabernaemontana divaricata* of family *Apocynaceae*.

**Figure 3 molecules-25-01904-f003:**
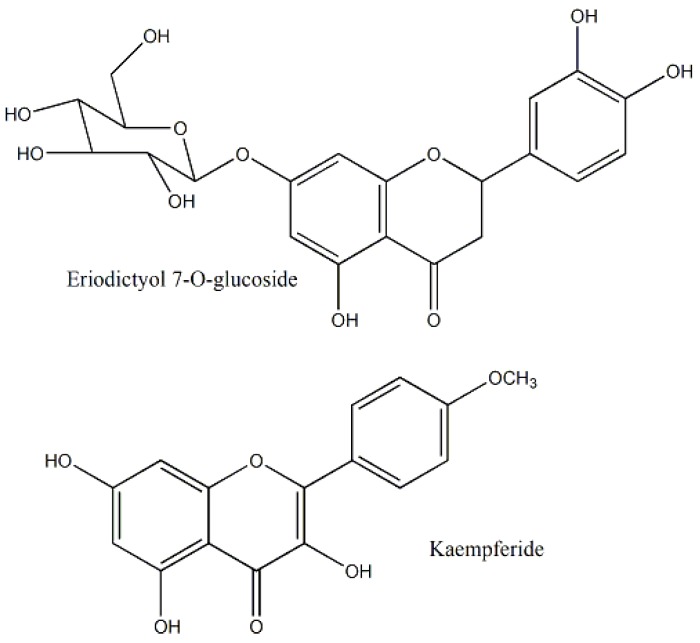
Cytotoxic constituents of *Luffa cylindrica* of family Araceae.

**Figure 4 molecules-25-01904-f004:**
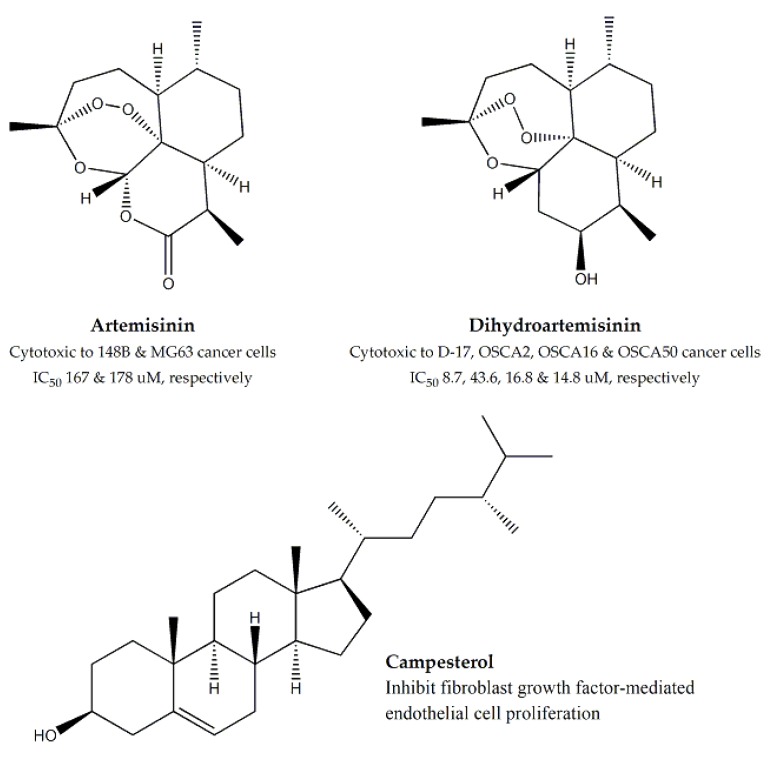
Cytotoxic constituent from family *Asterace.*

**Figure 5 molecules-25-01904-f005:**
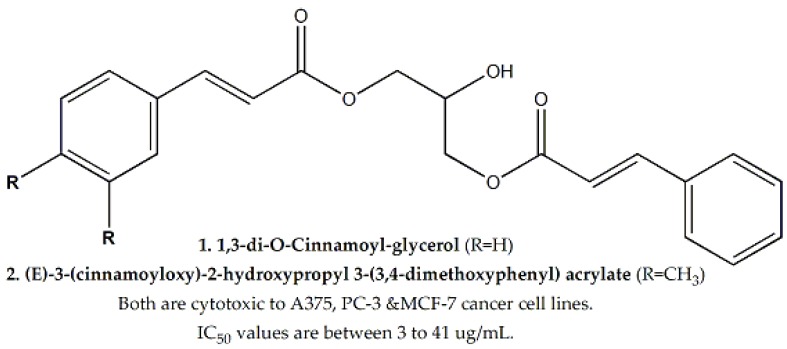
Bioactive dicinnamates from *Tillandsia recurvata* with cytotoxic potential.

**Figure 6 molecules-25-01904-f006:**
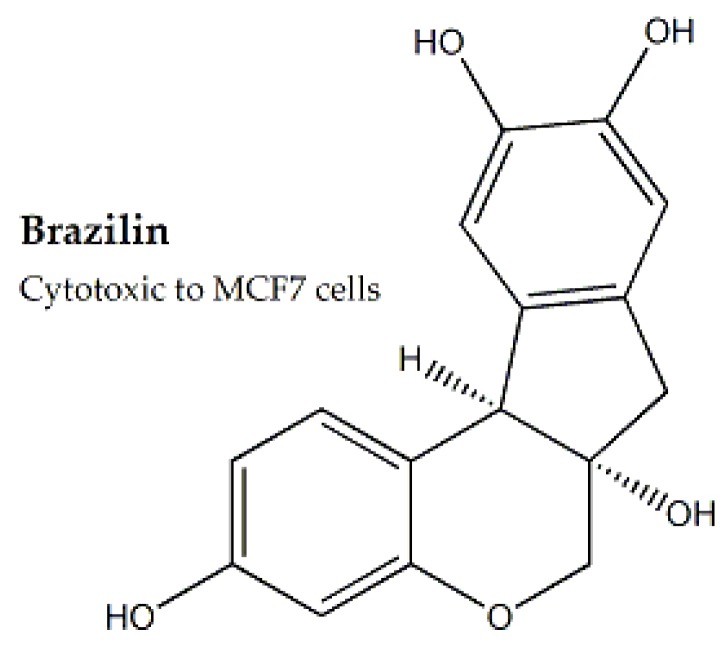
Cytotoxic Brazilin from *Caesalpinia sappan* heartwood.

**Figure 7 molecules-25-01904-f007:**
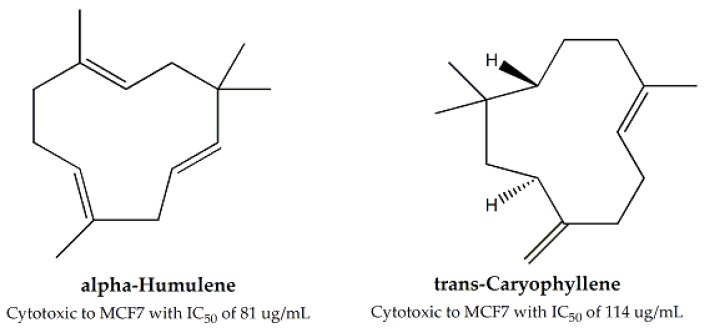
Cytotoxic components of family Lamiaceae.

**Figure 8 molecules-25-01904-f008:**
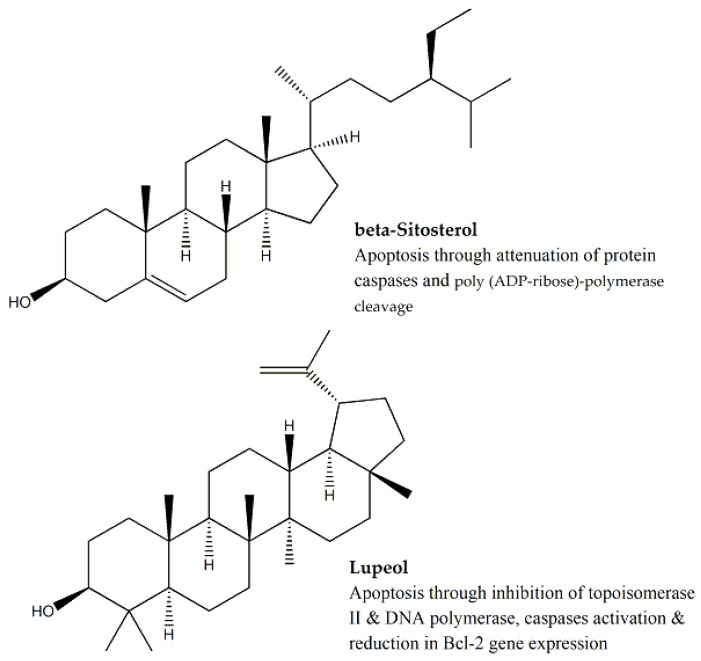
Biomarkers of family *Malvaceae* and mechanism of cytotoxicity.

**Figure 9 molecules-25-01904-f009:**
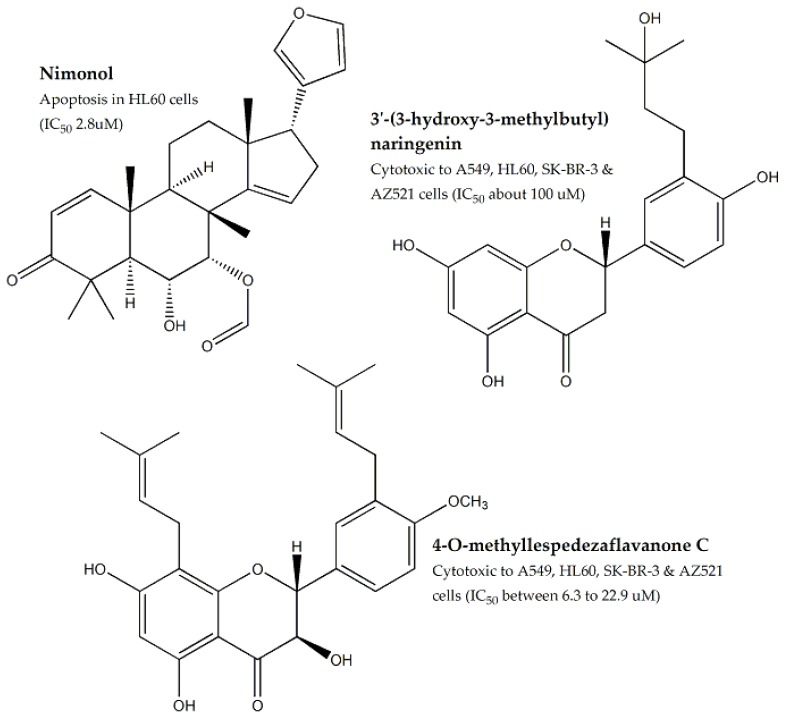
Cytotoxic components of *Azadirachta indica*.

**Figure 10 molecules-25-01904-f010:**
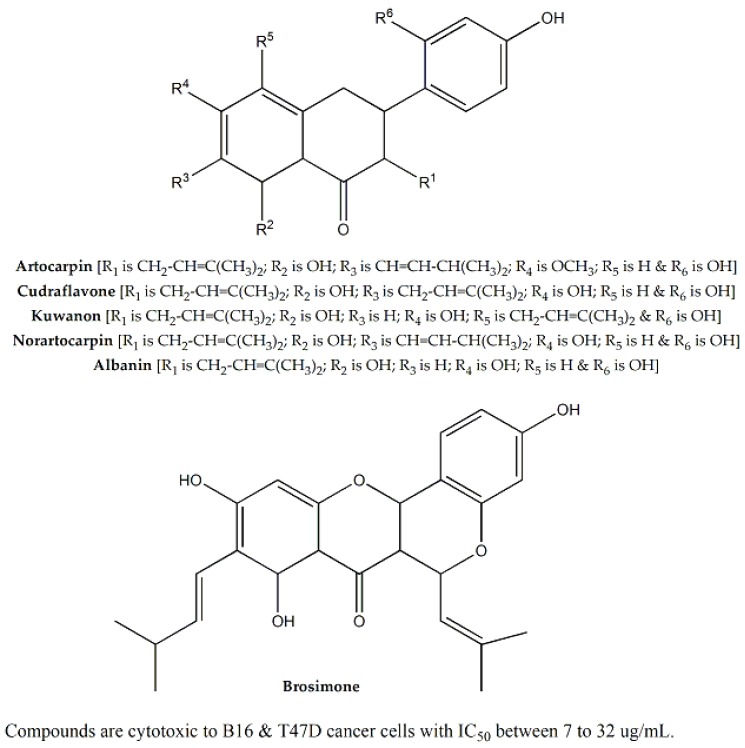
Cytotoxic phytoconstituents of family *Moraceae*.

**Figure 11 molecules-25-01904-f011:**
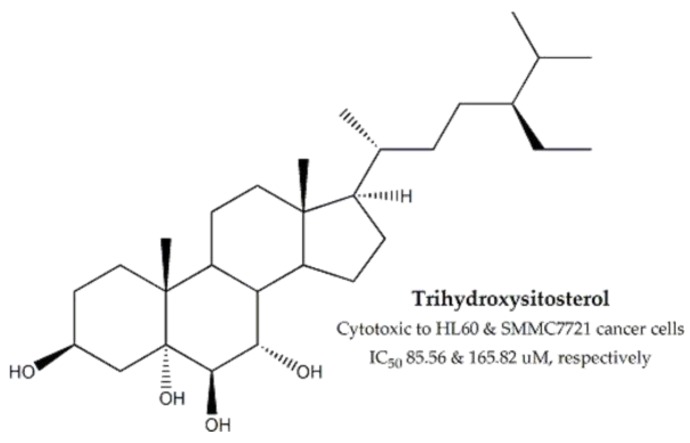
Bioactive cytotoxic component of *Phyllanthus emblica*.

**Figure 12 molecules-25-01904-f012:**
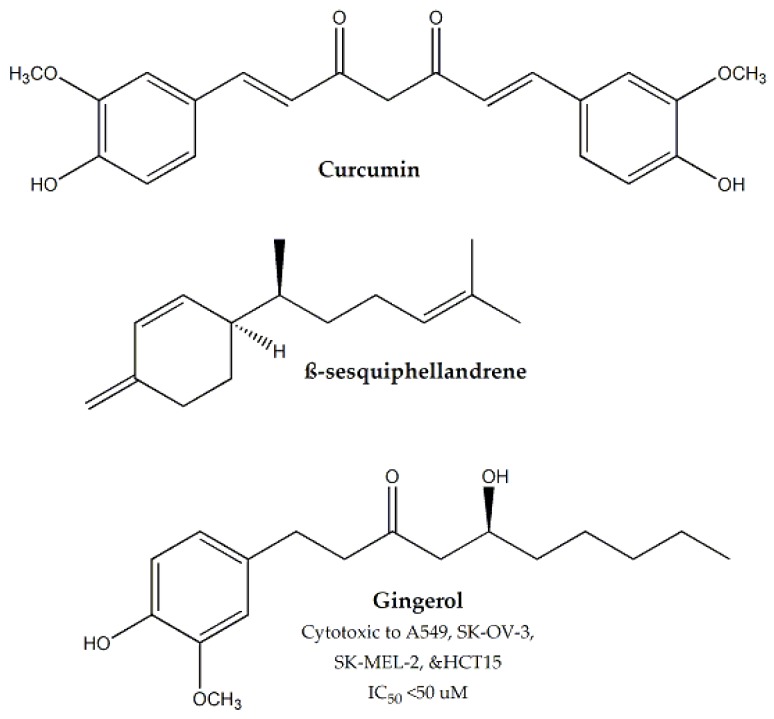
Cytotoxic compounds isolated from plants of family Zingiberaceae.

**Figure 13 molecules-25-01904-f013:**
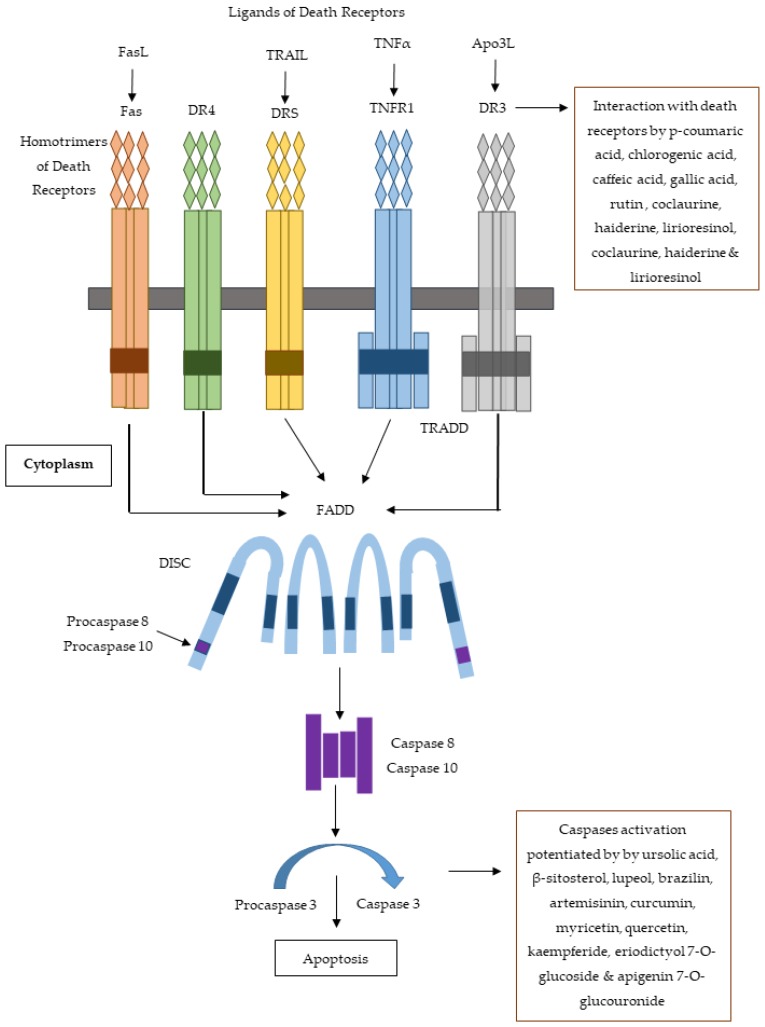
Extrinsic apoptosis pathway showing the sites of action of cytotoxic constituents obtained from medicinal plants of Indian subcontinent.

**Figure 14 molecules-25-01904-f014:**
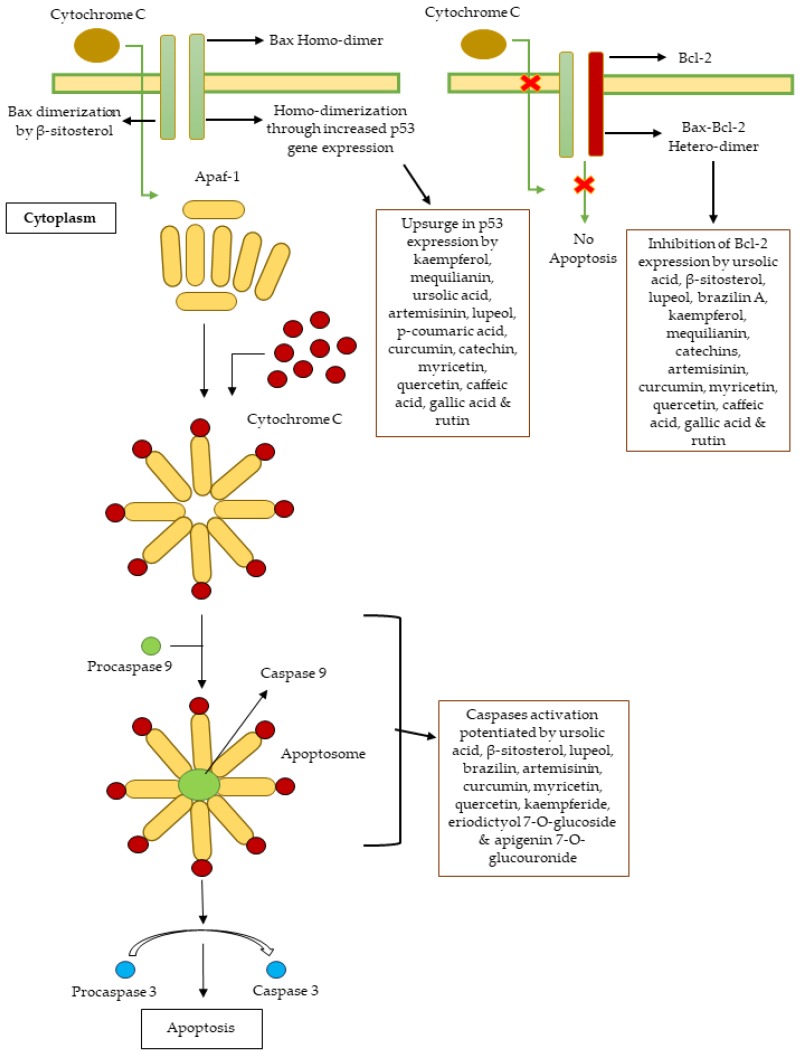
Intrinsic apoptosis pathway with sites of action of cytotoxic constituents obtained from medicinal plants of Indian subcontinent.

**Table 1 molecules-25-01904-t001:** Reported cytotoxic family of plants as well as their effectiveness against particular cell lines with quantification and isolated bioactive components.

Family	Genus and Species	Parts Used	Study Methods	Cell Line	IC_50_ (μg/mL)	Bioactive Compounds	IC_50_ (μg/mL) for Bioactive Compound	Mechanism of Cytotoxicity	Reference
**Acanthaceae**	*Barleria grandiflora*	Leaves	In vivo DAL induced ascitic tumor model in mice	DLA, A-549 and Vero	-	-	-	Upsurge in ascitic fluid	[17,18]
**Actinidiaceae**	*Saurauja roxburghii*	Leaves	MTT assay	A431 and C6 rat glioma	-	-	-	Interaction with caspases 3 and 9 topoisomerase I and p21^WAF1^ cell-cycle regulator	[20,21]
**Amaryllidaceae**	*Narcissus tazetta*	Aerial parts, Flowers	MTT assay	MCF-7 and Hep-2	-	-	-	-	[22]
**Anacardiaceae**	*Cotinus coggygria*	Leaves	MTT assay	HeLa and Vero	293 for HeLa	-	-	-	[23]
	*Lannea coromandelica*	Barks, Leaves	MTT assay	AGS, HT-29, MCF-7 and MDA-MB-231	90, 520, 270 and 160, respectively	2-Palmitoylglycerol, myricadiol, pyrogallol, isovanillin etc.	-	Apoptosis and DNA fragmentation	[24,25,26]
**Annonaceae**	*Annona muricata*	Leaves	MTT assay	EACC, MDA and SKBR3	335, 248 and 202, respectively	-	-	-	[27]
	*Annona squamosa*	Flowers	MTT assay	MCF-7 and Vero	6.9 and 75, respectively	Eupafolin, apigenin, rhamnetin etc.	-	-	[28]
**Apocynaceae**	*Tabernaemontana divaricata*	Flowers	MTT assay	HL-60, SMMC-7721, A-549, MCF-7 and SW480, NIH3T3 and HeLa	35.3 and >100 for NIH3T3 and HeLa, respectively	Ervachinine, cononitarine, conofoline, conophylline and cisplatin	Ranging between 0.17 to 1.49 for conophylline	-	[29,30]
**Araceae**	*Luffa cylindrica*	Aerial Parts	MTT assay	MCF-7 and Hep-2, BT-474, MDA-MB-231 and HepG2	329.2	Kaempferide, eriodictyol 7-*O*-glucoside and apigenin 7-*O*-glucouronide	-	Interaction with caspases 3 and 8	[22,31,32]
**Aristolochiaceae**	*Aristolochia ringens*	Roots	*In vivo* S 180 induced ascetic and solid tumor model in mice	HeLa, A-431, A-549, PC-3, HCT-116 and THP-1	Ranging between 3 to 24	**-**	**-**	**-**	[33]
	*Aristolochia longa*	Roots	MTT assay	MCF-7, HT-29, H5-6 and N2A	-	-	-	-	[34]
**Asphodelaceae**	*Aloe vera*	Whole Plant	MTT assay	HepG2	10.5	-	-	Increased *p53* and reduced *Bcl-2* gene expression	[35]
**Aspleniaceae**	*Asplenium nidus*	Whole Plant	MTT assay	HepG2 and HeLa	-	Gliricidin-7-*O*- hexoside	About 507	-	[36]
**Asteraceae**	*Artemisia annua*	Aerial Parts	MTT assay	148B, MG63, D-17, OSCA2, OSCA16 and OSCA50	167 and 178 for 148B and MG63, respectively	Dihydro- artemisinin	Ranging between 8.7 to 43.6	-	[37,38,39,40]
	*Bidens pilosa*	Whole Plant	MTT and Comet assays	HeLa and KB	88.6 and 372.4, respectively	-	-	-	[41]
	*Centaurea antiochia*	Aerial Parts	MTT assay	Vero and HeLa	-	-	-	-	[23]
	*Centaurea nerimaniae*	Aerial Parts	MTT assay	Vero and HeLa	-	-	-	-	[23,42]
	*Chrysanthemum coronarium*	Aerial Parts	*In vitro* MTT assay and in vivo chorioallantoic membrane (CAM) model	MCF-7, T47D, CACO-2, HRT18, A375.S2 and WM1361A	75.7, 79.8 and 138.5 for WM1361A, T47D and MCF-7	Campesterol	-	Inhibition of fibroblast growth factor-mediated cell proliferation	[43,44]
	*Inula viscosa*	Flowers	MTT assay	MCF-7 and Hep-2	15.78 for MCF-7	-	-	-	[22]
	*Parthenium hysterophorus*	Leaves	MTT and SRB assays	DU-145, THP-1, MCF-7 and K562	-	-	-	-	[42,45]
	*Phagnalon rupestre*	Aerial Parts	MTT assay	MCF-7 and Hep-2	93.3 and 97, respectively	-	-	-	[22]
	*Scorzonera tomentosa*	Aerial Parts	MTT assay	Vero and HeLa	987 and 195, respectively	**-**	**-**	**-**	[23]
	*Senecio scandens*	Leaves	MTT and Clonogenic assays	DL, MCF-7 and HeLa	27, 24 and 18, respectively	-	-	-	[46]
**Berberidaceae**	*Berberis aristata*	Roots, Stem	MTT assay	MCF-7, DWD, Hop62 and A2780	71 and 220 for MCF-7 and DWD	-	-	-	[47,48]
**Bignoniaceae**	*Tecoma stans*	Leaves, Flowers	MTT assay	A549 and HepG2	-	Rutin, luteolin, diosmetin, skytanthine, etc.	-	-	[49,50,51]
**Boraginaceae**	*Cordia dichotoma*	Leaves	MTT, DCFH-DA and DAPI staining assays	PC3	74.5	-	-	apoptosis, nuclear condensation and ROS production	[52]
**Bromeliaceae**	*Tillandsia recurvata*	Whole Plant	WST-1 assay	A375, MCF-7 and PC-3	1, 40.5 and 6, respectively	1,3-di-*O*-Cinnamoyl-glycerol and (*E*)-3-(cinnamoyloxy)-2-hydroxypropyl 3-(3,4-dimethoxyphenyl) acrylate	Ranging between 3 to 41	-	[53]
**Caesalpiniaceae**	*Caesalpinia sappan*	Heartwood, Leaves	MTT assay	MCF-7 and A-549	-	Brazilin A	-	Reduction of *Bcl-2* apoptotic inhibitor	[54]
	*Saraca asoca*	Leaves	MTT assay	AGS	20	-	-	-	[24]
**Compositae**	*Gnaphalium luteoalbum*	Leaves	MTT assay	VERO, NIH3T3, AGS, HT-29, MCF-7 and MDA-MB-231	980 and 340 for AGS and MCF-7, respectively	Apigenin, luteolin, jaceosidin, gnaphalin, etc.	-	-	[24,55]
**Dilleniaceae**	*Dillenia pentagyna*	Stem Bark	*In vivo* DAL induced ascitic tumor model in mice	DL, MCF-7 and HeLa	25.8, 41.6 and 76.8, respectively	-	-	Reduction in glutathione level	[46,56]
	*Dillenia indica*	Leaves	MTT assay	MCF-7 and MDA-MB-231	340 and 540, respectively	-	-	-	[24]
**Dipsacaceae**	*Pterocephalus pulverulentus*	Aerial Parts	MTT assay	MCF-7 and Hep-2	-	-	-	-	[22]
**Ebenaceae**	*Diospyros peregrina*	Leaves, Fruits	MTT assay	MCF-7, HepG2 and MDA-MB-231	7, 64.4 and 33, respectively	-	-	-	[24,57]
**Ericaceae**	*Arbutus andrachne*	Aerial Parts	MTT assay	MCF-7, T47D, CACO-2, HRT-18, A375.S2 and WM1361A	-	-	-	-	[43]
**Euphorbiaceae**	*Croton caudatus*	Leaves	MTT and Clonogenic assays	DL, MCF-7 and HeLa	29.7 for DL	-	-	-	[46]
	*Euphorbia tirucalli*	Leaves, Stem	*In vitro* cell viability study with CCK-8	MiaPaCa-2	-	-	-	-	[58]
**Fabaceae**	*Adenanthera pavonina*	Stem Bark	*In vivo* DAL induced ascitic tumor model in mice	DAL	-	-	-	-	[18]
	*Ononis hirta*	Aerial Parts	MTT assay	MCF-7, Hep-2 and Vero	28, 48.8 and 41.9, respectively	-	-	-	[22]
	*Ononis sicula*	Aerial Parts	MTT assay	MCF-7, Hep-2 and Vero	66, 75.3 and 79.5, respectively	-	-	-	[22]
**Gramineae**	*Zea mays*	Leaves	MTT and SRB assays	Hep-2, A549, SK-OV-3, SK-MEL-2, XF-489 and HCT-15	-	Maysin	Ranging between 32 to 62	-	[59,60]
**Hypericaceae**	*Hypericum kotschyanum*	Aerial Parts	MTT assay	HeLa and Vero	507 and 367, respectively	-	-	-	[23]
**Labiatae**	*Salvia pinardi*	Aerial Parts	MTT assay	MCF-7 and Hep-2	85.5 for MCF-7	-	-	-	[22]
**Lamiaceae**	*Lavandula angustifolia*	Flowers	MTT assay	MCF-7 and Hep-2	-	-	-	-	[22]
	*Nepeta italica*	Aerial Parts	MTT assay	HeLa and Vero	980 and >1000, respectively	-	-	-	[23]
	*Plectranthus stocksii*	Leaves, Stem	MTT assay	RAW264.7, Caco-2 and MCF-7	9, 36.1 and 48.9, respectively	-	-	-	[61]
	*Salvia officinalis*	Leaves, Stem	“Alamar Blue” resazurin reduction assays	MCF-7, B16F10 and HeLa	Ranging between 14 – 36	α-Humulene and *trans*-caryophyllene	For MCF-7, 81 and 114, respectively	-	[62]
	*Ocimum sanctum*	Leaves	*In vivo* S 180 induced mice model	DMBA-induced hamster and S-180 induced mice	-	-	-	Regulation of humoral immunity and stimulation of cell mediated immunity	[63,64,65,66]
	*Origanum sipyleum*	Aerial Parts	MTT assay	HeLa and Vero	>1000	-	-	-	[23]
	*Salvia hypargeia*	Aerial Parts	MTT assay	Vero and HeLa	> 1000	-	-	-	[23]
	*Teucrium polium*	Aerial Parts	MTT assay	MCF-7, Hep-2, T47D, CACO-2, HRT18, A375.S2 and WM1361A	-	-	-	-	[22,43]
	*Teucrium sandrasicum*	Aerial Parts	MTT assay	HeLa and Vero	513 and 593, respectively	-	-	-	[23]
**Malvaceae**	*Hibiscus calyphyllus*	Aerial Parts	MTT assay	HepG2 and MCF-7	14.5 and 25.1, respectively	Ursolic acid, β-sitosterol and lupeol	-	Interaction with caspases 3 and 9, topoisomerase I, *Bcl-2*, *Bax*, DNA polymerase and poly (ADP-ribose)-polymerase	[67,68,69,70]
	*Hibiscus deflersii*	Aerial Parts	MTT assay	HepG2 and MCF-7	14.4 and 11.1, respectively	Ursolic acid, β-sitosterol and lupeol	-	Interaction with caspases 3 and 9, topoisomerase I, *Bcl-2*, *Bax*, DNA polymerase and poly (ADP-ribose)-polymerase	[67,68,69,70]
	*Hibiscus micranthus*	Aerial Parts	MTT assay	HepG2 and MCF-7	27.6 and 24.1, respectively	Ursolic acid, β-sitosterol and lupeol	-	Interaction with caspases 3 and 9, topoisomerase I, *Bcl-2*, *Bax*, DNA polymerase and poly (ADP-ribose)-polymerase	[67,68,69,70]
**Meliaceae**	*Azadirachta indica*	Leaves, Seeds	MTT assay	HT-29, A-549, MCF-7, HepG-2, MDBK, EAC, HL60, SK-BR-3, AZ521, SW-480, and SMMC7721	Ranging between 83.5 to 212.2	Nimonol, 3′-(3-hydroxy-3-methyl-butyl)naringenin, 4′-*O*-methyl- lespedezaflavanone C, triterpenoids, limonoids etc.	Ranging between 4.2 to 100	-	[71,72,73,74,75,76,77]
**Menispermaceae**	*Cocculus hirsutus*	Aerial Parts	MTT assay	MCF-7	39.1	Coclaurine, haiderine, lirioresinol etc.	-	Interactions with Aurora kinase, c-Kit, FGF, *NF-kB*, *Bcl-xL* and VEGF	[78,79]
**Moraceae**	*Artocarpus heterophyllus*	Seeds	SRB assay and MTT assays	HEK293, A549, HeLa, MCF-7, B16 and T47D	35.3 for A549	Noratocarpin, cudraflavone, artocarpin, brosimone, kuwanon, albanin, etc.	Ranging between 7 to 32	Inhibition of melanin biosynthesis	[80,81,82,83]
	*Ficus beecheyana*	Roots	MTT assay	AGS, SW-872, HL-60 and HepG2	-	*p*-Coumaric acid, chlorogenic acid, caffeic acid, gallic acid, rutin etc.	-	Interaction with Fas, Fas-L, *p53*, *Bcl-2* and caspases (3,8,9) proteins	[84]
	*Ficus carica*	Fruits, Leaves	“Alamar Blue” resazurin reduction assays	MCF-7, B16F10 and HeLa	440, 880 and >1000, respectively	-	-	-	[62]
	*Ficus racemosa*	Fruits	SRB assay	MCF-7, DLA, HL-60, HepG2, NCI-H23 and HEK-293T	80, 175, 276.9 and 363 for MCF-7, DLA, HL-60 and HepG2, respectively	-	-	-	[85,86,87]
	*Morus nigra*	Leaves	MTT assay	OVCAR-8, SF-295, HCT-116 and HeLa	-	-	-	-	[88,89]
**Myristicaceae**	*Myristica fragrans*	Barks	MTT assay	KB, K-562, HCT-116 and MCF-7	75 for KB	38 Essential oils	Ranging between 2.11 – 78.15	Apoptosis through *Bcl-2* inhibition	[90,91]
**Myrtaceae**	*Syzygium cumini*	Seeds	MTS cell proliferation assay	A2780, MCF-7, PC3 and H460	49, 110, 140 and 165, respectively	Pelargonidin-3-*O*- glucoside, cyanidin-3-*O*-malonyl glucoside, delphenidin-3-*O*- glucoside, ellagitannins etc.	-	-	[92,93,94]
**Nyctaginaceae**	*Mirabilis jalapa*	Aerial Parts, Roots, Stem	MTT assay	HeLa, Raji, A549, HCT 116, Vero MCF-7 and Hep-2	-	Rotenoids and mirabilis antiviral protein (MAP)	For MAP, 150, 175, 200 against HCT116, MCF-7 and A549, respectively	-	[22,95,96]
**Oleaceae**	*Jasminum sambac*	Flowers, Leaves	MTT assay	MCF-7 and HEP-2	7 for MCF-7	-	-	-	[22,24]
	*Olea europaea*	Leaves	MTT assay	MCF-7, B16F10, HeLa and Vero	43, 170, 440 and >1000, respectively			Interaction with *Bcl-2*, *Bax*, and *p53* proteins	[23,62,97]
	*Syringa vulgaris*	Fruits	MTT assay	MCF-7 and HeLa	-	-	-	-	[22]
**Oxalidaceae**	*Averrhoa bilimbi*	Fruits, Leaves	MTT assay	MCF-7	154.9 and 668 for fruits and leaves	Nonanal, tricosane, squalene, malonic acid, etc.	-	-	[98,99]
**Phyllanthacae**	*Phyllanthus emblica*	Fruits, Leaves	MTT assay	HT-29, HL-60 and SMMC-7721	35 for HT-29	Trihydroxy- sitosterol	85.56 and 165.82 for HL-60 and SMMC-7721, respectively	-	[100,101,102]
**Plumbaginaceae**	*Limonium densiflorum*	Shoots	Resazurin reduction assays	A-549 and DLD-1	29 and 85, respectively	Myricetin, isorhamnetin and *trans*- 3-hydroxy- cinnamic acid	-	-	[103]
**Picrorhiza**	*Picrorhiza kurroa*	Rhizomes	XTT and SRB assays	Hep3B, MDA-MB-43 and PC-3	32.6, 19 and 63.9, respectively	Curcubitacin	-	Apoptosis	[104,105]
**Pinaceae**	*Cedrus deodara*	Woods	SRB assay	MIA-PA-CA-2, PC3, A-2780 and Molt-4	74, 77, 63 and 15, respectively	-	-	Interaction with caspase 3,8 and 9 proteins	[48,106]
**Piperaceae**	*Piper longum*	Fruits	*In vitro* SRB assay, in vivo DLA and EAC induced ascites tumor model in mice	Colo-205, DLA and EAC	18 for Colo-205	Piperine	-	Apoptosis	[48,107]
	*Piper regnellii*	Leaves	MTT assay	UACC-62, 786–0, MCF-7, NCI-H460, OVCAR-3, PC-3, HT-29 and K-562	-	Eupomatenoid-5	-	-	[108]
**Poaceae**	*Cenchrus ciliaris*	Aerial Parts, Roots	MTT assay	A-549, CACO, HCT-116, HeLa, HepG2, MCF-7 and PC3	Ranging between 9 to 27.2	-	-	-	[109,110]
**Polygonaceae**	*Calligonum comosum*	Whole Plant	MTT assay	HepG2	9.6	Catechin, kaempferol, mequilianin, etc.	-	Increased *p53* and reduced *Bcl-2* gene expression	[35,111]
**Primulaceae**	*Aegiceras corniculatum*	Fruits	MTT assay	VERO, NIH3T3, AGS HT-29, MCF-7 and MDA-MB-231	150, 97, 0.5, 998, 91 and 461, respectively	Aegicoroside A, sakurasosaponin, fusarine, fusamine etc.	-	-	[24,112,113]
**Ranunculaceae**	*Delphinium staphisagria*	Seeds	MTT assay	MCF7, HT29, N2A, H5-6 and VCREMS	41.9, 41.1 and 14.5 for N2A, H5-6 and VCREMS, respectively	Astragalin, paeonoside, petiolaroside, etc.	-	-	[34,114]
**Rosaceae**	*Crataegus microphylla*	Leaves, Flowers	MTT assay	HeLa and Vero	576 for HeLa	-	-	-	[23,115]
	*Rosa damascena*	Flowers, Seeds	MTT assay	MCF-7, Hep-2, HeLa and Vero	265 for HeLa	Nerol, geraniol, β-citronellol, linalool, nonadecane and phenylethyl alcohol	-	-	[22,23,116]
**Rubiaceae**	*Hymenodictyon excelsum*	Barks, Woods	MTT assay	VERO, NIH3T3, AGS HT-29, MCF-7 and MDA-MB-231	230, 70, 90, 160, 80 and 440, respectively	-	-	DNA fragmentation and apoptosis	[24,117]
	*Oldenlandia corymbosa*	Leaves	MTT assay	K562	114.4	-	-	Apoptosis	[118]
**Salicaceae**	*Populus alba*	Flowers	MTT assay	Hep-2, A549, H1299 and MCF-7	12.05, 10.53 and 28.16 for A549, H1299 and MCF-7	-	-	-	[22,119]
**Sapotaceae**	*Manilkara zapota*	Flowers	MTT assay	MCF-7, HT-29 and HL-60	12.5	Lupanes, oleananes, ursanes, manilkoraside, etc.	For manilkoraside, 64 and 24 against HT-29 and HL-60, respectively	DNA fragmentation	[28,120]
**Saururaceae**	*Saururus chinensis*	Roots	MTT assay	MCF-7, HT-29 and HepG2	91.2	Aristolactram, dihydroguaiatric acid, sauchinone, saucerneol D, manassantin A and B, saucerneol F etc.	Ranging between 10-16	Inhibition of DNA topoisomerase I and II	[121,122]
**Scrophulariaceae**	*Verbascum sinaiticum*	Aerial Parts, Flowers	MTT assay	MCF-7 and Hep-2		Sinaiticin, hydrocarpin and flavonolignans	For hydrocarpin and sinaiticin, 1.2 and 7.7 against P-388	-	[22,123]
**Solanaceae**	*Solanum khasianum*	Fruits	MTT and Clonogenic assays	DL, MCF-7 and HeLa	27.4, 71.2 and 62.5, respectively	-	-	-	[46]
	*Solanum nigrum*	Fruits	MTT and Clonogenic assays	HeLa, Vero, HepG2 and CT26	265, 6.9, 56.4 and 77.6, respectively	-	-	-	[124,125]
	*Withania coagulans*	Fruits, Roots and Leaves	Presto Blue cell viability assay	HeLa, MCF-7 and RD	Ranging between 0.7 to 6.7	Myricetin, quercetin, gallic acid, etc.	-	-	[126]
	*Withania somnifera*	Roots	SRB assay	PC3, A-549, A-2780 and K-562	52, 46, 79 and 41, respectively	-	-	-	[48]
**Sterculiaceae**	*Helicteres isora*	Whole Plant	MTT assay	HeLa-B75, HL-60, HEP-3B and PN-15	-	Cucurbitacin B and isocucurbitacin B	-	-	[127,128]
**Thymelaeaceae**	*Aquilaria malaccensis*	Leaves	*In vitro* TBE and MTT assay, in vivo EAC induced ascites model	HCT116, DLA and EAC	4, 72 and 79, respectively	Benzaldehyde, pinene, octanol, germacrene, hexadecanal, etc.	-	-	[129,130,131]
**Verbenaceae**	*Clerodendrum viscosum*	Leaves	MTT assay	VERO, NIH3T3, AGS, HT-29, MCF-7 and MDA-MB-231	50 and 880 for MCF-7 and HT-29	-	-	Apoptosis	[24]
	*Clerodendron infortunatum*	Roots	DLA induced ascites tumor model in mice	DLA	-	-	-	Apoptosis through interaction with *Bax*, *Bcl-2*, caspases 8 and 10 proteins	[132]
**Vitaceae**	*Leea indica*	Leaves	MTT assay	DU-145 and PC-3	529.4 and 677.1, respectively	-	-	-	[133]
	*Vitis vinifera*	Stem	“Alamar Blue” resazurin reduction assays	MCF-7, B16F10 and HeLa	62, 137 and 336, respectively	Quercetin 3-*O*-β-D-4C_1_ galactoside and quercetin 3-*O*-β-D-4C_1_ glucuronide	-	Apoptosis	[62,134,135]
**Zingiberaceae**	*Curcuma longa*	Rhizomes	MTT assay	Hep-2	-	Curcumin, β-sesquiphellandrene	-	Apoptosis	[136,137,138,139,140]
	*Zingiber officinale*	Rhizomes	*In vitro* MTT assay, in vivo DLA and EAC induced ascites model	DLA, EAC, A549, SK-OV-3, SK-MEL-2, PC-3M and HCT15	-	Gingerol	< 50	-	[140,141]

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
