# Peer review of "A Review of Cytotoxic Plants of the Indian Subcontinent and a Broad-Spectrum Analysis of Their Bioactive Compounds"

_molecules, 2020, doi:10.3390/molecules25081904_

Round 1

Reviewer 1 Report

The  paper describes well anti-cancer compounds  isolated from Indian plants.

Author Response

Dear Reviewer,

We express our sincere  gratitude for your kind review and comments.

Reviewer 2 Report

The authors of this paper aimed at scanning the scientific literature in a search for bioactive compounds from medicinal plants of the Indian subcontinent, which may turn to be of potential usefulness in the therapy of cancer in humans. They reviewed the observations collected in vitro with 98 plant species from 57 families, and with 80 isolated compounds from the same starting species, mostly by the use of such assays as the widely used enzymatic reduction of [3-(4,5-dimethylthiazol-2-yl)-2,5-diphenyl-tetrazolium bromide] (MTT assay) or similar ones. All of the plant species investigated were from the Indian subcontinent.

Major issues

Methodology:

- A methodological section is lacking in this paper. The author should explain how the botanical families, genera and species reviewed in this manuscript were selected. Did the authors make a systematic search of the literature? Which databases did they scan? Which key words did they use? What inclusion and exclusion criteria did they adopt, if any?

- Moreover, for many years now, a number of resources to facilitate the discovery and development of new cancer therapeutic agents have been developed and made available to the world community of scientists by large scientific Institutions, such as the Developmental Therapeutic Program (DTP) of the National Cancer Institute at the National Institutes of Health, USA, including DTP databases and searching/analysis tools, and DTP data repositories for natural products. Did the authors consult or refer to any of these large collections of data, concerning the anticancer properties of compounds of botanical sources? Whether they did or not, which were the reason(s) why? In case they did, what criteria did they use, other than a “local” perspective, in selecting the plants and compounds covered in the manuscript?

- As a matter of fact, all of the 98 source plant species covered in this review were from the Indian subcontinent. Although 98 may seem a large number at first sight, it doesn’t stand the consideration that a couple of thousand novel botanical species are uncovered each year in the ongoing effort of counting and naming the world’s plants. As for the “local” approach, it appears to me as a weakness of this review. I deem it that practical and echonomical considerations, that may prompt scientist to investigate “in-house” natural compounds, may in fact turn out to be impractical, not only from a scientific, but also from an echonomical standpoint, should they favour the development and adoption of “local” bioactive compounds of modest potency, over more active compounds of “foreign” origin.

Data report and analysis:

- This entire manuscript is more like a draft, containing only a list of plant families, genera and species, with preliminary notes on some biological effects observed, mainly in the context of in vitro cell assays. Some of the plants and their active compounds are covered in more detail, whilst the majority of them is mentioned and described very briefly.

- In page 1, line 22 of the Abstract the authors state that their “review supports further investigation of subcontinent medicinal plants as an important source of new drug leads”. Now, before a compound may be defined a “lead” for the development of an active principle of any preventive or even therapeutic usefulness, it is essential that animal and pre-clinical studies be conducted, addressing quantitative aspects of its pharmacodynamics, such as the amount and rate of absorption in the gastrointestinal tract, its biotransformation in liver, its rate of appearance, concentration attainable and half-life in blood and its biodistribution to peripheral organs. The complete lack of any pharmacokinetic data is another major point of weakness of the present review. The authors are urged to include these aspect in the scope of their manuscript, in order to confer some more usefulness upon the effort of reading their paper. In the meantime, they should abstain from referring to the compounds listed in their review as to potential “new drug leads”.

- As for the conclusions drawn by the authors, no critical issues are raised on the studies which were examined in this review, in relation either with the methodologies used, the results obtained, or the conclusions drawn. No hints are derived from the collection of the different studies. No implications are suggested, nor are prompts for further investigation. As outlined above, a critical and collective discussion of the data reported is totally lacking.

- Tables 1 and 2 are partially duplicates of each other. As a matter of fact, all of the source plants listed in Table 2 are also listed in Table 1. Thus, the sole additional information that the latter provides concerns the plant parts used and the assay types employed. As both tables are exceedingly long, they ought to be combined in one comprehensive table.

- Linguistic revision: the manuscript is in need of a linguistic revision for frequent mistakes in English vocabulary, syntax and punctuation throughout. Taxonomic names aren’t exempt from mistakes and inconsistencies.

Minor points

- Page 1, lines 30-37: this explanation of what cancer is and which its etiologic and pathogenetic factors are is both unneeded and too generical, or frankly misleading. As an instance, it is not correct that “unrestrained proliferation of a normal cell often produces genetic alterations and instabilities which accumulates within tissues and cells”, but it seems to be rather the opposite. Furthermore, what do the authors mean by “hormonal immune system?”. And further, references 1 through 3 are inappropriate. This entire section of the introduction should be simply deleted.

- Abbreviations: such acronyms as DAL, MTT, DCFH-DA, DAPI, WST-1, CCK-8, SRB, MTS,  XTT ought to be spelled out the first time they appear in the text body, and a list of abbreviations should be provided as an addition to it.

- Tables 1 and 2, column 2: please replace “Plant” with “Genus and species”.

Author Response

Dear Reviewer,

Thank you very much  for your kind review and necessary comments. Please find our response below:

Major issues

Comment 01:  A methodological section is lacking in this paper. The author should explain how the botanical families, genera and species reviewed in this manuscript were selected. Did the authors make a systematic search of the literature? Which databases did they scan? Which key words did they use? What inclusion and exclusion criteria did they adopt, if any?

Response 01: A methodological section has been added to the manuscript with description of the databases used, search key words, as well as the inclusion and exclusion criteria.

Comment 02: Moreover, for many years now, a number of resources to facilitate the discovery and development of new cancer therapeutic agents have been developed and made available to the world community of scientists by large scientific Institutions, such as the Developmental Therapeutic Program (DTP) of the National Cancer Institute at the National Institutes of Health, USA, including DTP databases and searching/analysis tools, and DTP data repositories for natural products. Did the authors consult or refer to any database of these large collections of data, concerning the anticancer properties of compounds of botanical sources? Whether they did or not, which were the reason(s) why? In case they did, what criteria did they use, other than a “local” perspective, in selecting the plants and compounds covered in the manuscript?

Response 02: The cytotoxic compounds mentioned in this review have further evaluated using the DTP database. Using CAS No. of the compounds, the authors have searched the DTP database and information on in-vivo studies as well as reports on pharmacokinetic and clinical trials have been included in the review.

Comment 03: As for the “local” approach, it appears to as a weakness of this review.

Response 03: The authors attempted to signify the contribution of Indian subcontinent, known as “the botanical garden of the world”, in the arena of drug discovery and development. The authors also tried to highlight the insignificant amount of works that have been done on the plants of the region as well as to amplify the importance of this area’s natural resources to the scientific community.

Data report and analysis:

Comment 04: This entire manuscript is more like a draft, containing only a list of plant families, genera and species, with preliminary notes on some biological effects observed, mainly in the context of in vitro cell assays. Some of the plants and their active compounds are covered in more detail, whilst the majority of them is mentioned and described very briefly.

Response 04: The authors further have tried to include more information on the families that have been described in brief in the report. Although, due to the insignificant amount of works, not a lot of information could have been compiled.

Comment 05:  In page 1, line 22 of the Abstract, the authors stated that their “review supports further investigation of subcontinent medicinal plants as an important source of new drug leads”. Now, before a compound may be defined a “lead” for the development of an active principle of any preventive or even therapeutic usefulness, it is essential that animal and pre-clinical studies be conducted, addressing quantitative aspects of its pharmacodynamics, such as the amount and rate of absorption in the gastrointestinal tract, its biotransformation in liver, its rate of appearance, concentration attainable and half-life in blood and its biodistribution to peripheral organs. The complete lack of any pharmacokinetic data is another major point of weakness of the present review. The authors are urged to include these aspect in the scope of their manuscript, in order to confer some more usefulness upon the effort of reading their paper. In the meantime, they should abstain from referring to the compounds listed in their review as to potential “new drug leads”.

Response 05: The few available pharmacokinetic data on handful of compounds have been stated in the review under a new section namely “Reported Cytotoxic Constituents: Therapeutic Perspective and Future Directions”. Most of the reported compounds have not yet been assayed in details. The authors also excluded words like “new drug leads”.

Comment 06:  As for the conclusions drawn by the authors, no critical issues are raised on the studies which were examined in this review, in relation either with the methodologies used, the results obtained, or the conclusions drawn. No hints are derived from the collection of the different studies. No implications are suggested, nor are prompts for further investigation. As outlined above, a critical and collective discussion of the data reported is totally lacking.

Response 06: A critical and collective discussion as well as future directions have been added to the review under the new section mentioned above.

Comment 07:  Tables 1 and 2 are partially duplicates of each other. As a matter of fact, all of the source plants listed in Table 2 are also listed in Table 1. Thus, the sole additional information that the latter provides concerns the plant parts used and the assay types employed. As both tables are exceedingly long, they ought to be combined in one comprehensive table.

Response 07: The tables have been combined in one comprehensive table.

Response 08: Linguistic revision: the manuscript is in need of a linguistic revision for frequent mistakes in English vocabulary, syntax and punctuation throughout. Taxonomic names aren’t exempt from mistakes and inconsistencies.

 Comment 08: A linguistic revision was carried out to correct mistakes in English vocabulary, syntax, punctuation and taxonomic errors.

Minor points

Comment 09: Page 1, lines 30-37: this explanation of what cancer is and which its etiologic and pathogenetic factors are is both unneeded and too generical, or frankly misleading. As an instance, it is not correct that “unrestrained proliferation of a normal cell often produces genetic alterations and instabilities which accumulates within tissues and cells”, but it seems to be rather the opposite. Furthermore, what do the authors mean by “hormonal immune system?” References 1 through 3 are inappropriate. This entire section of the introduction should be simply deleted.

Response 09: Lines 30 to 37 of “Introduction” section, including the explanation of what cancer is as well as mistakes like “hormonal immune system”, were reviewed and corrected extensively.

Comment 10: Abbreviations: such acronyms as DAL, MTT, DCFH-DA, DAPI, WST-1, CCK-8, SRB, MTS, XTT ought to be spelled out the first time they appear in the text body, and a list of abbreviations should be provided as an addition to it.

Response 10: An “Abbreviations” section has been added to the manuscript and the acronyms were spelled out the first time they appear in the text body.

Comment 11: Tables 1 and 2, column 2: please replace “Plant” with “Genus and species”.

Response 11: “Plant” was replaced with “Genus and species” in the table.

Reviewer 3 Report

The tables in this review are very useful to summarize the cytotoxic plants of the Indian subcontinent highlighting the bioactive compounds.

I have a few comments:

  1. There are some grammatical changes/edits that need to be made throughout. One example: line 18-19 – “Therefore, in present review, an attempt has been made to concise the reported with description of…” should say “Therefore, in the present review, an attempt has been made to concisely report the cytotoxic plants… with a description of…”

  1. How were all the chemical structures made? Using what software? Or taken from where?

  1. In the figure legend or in the figure if possible for figure 10 and 11 show all of the bioactive ingredients that interfere with the major parts of these pathways.

  1. This review is an excellent summary of the literature but it seems so descriptive. I believe a review should add analysis/interpretation and direction for the field. This is only a summary of journal articles.

  1. How were these articles chosen? How many exist versus how many were summarized here?

Author Response

Dear Reviewer,

Thank you so much for your kind review and necessary comments. Please find the response as below:

Comment 01: There are some grammatical changes/edits that need to be made throughout. One example: line 18-19 – “Therefore, in present review, an attempt has been made to concise the reported with description of…” should say “Therefore, in the present review, an attempt has been made to concisely report the cytotoxic plants… with a description of…”

Response 01: The grammatical changes/edits have been made throughout to the best of the authors’ knowledge which include the mistakes in lines 18-19.

Comment 02: How were all the chemical structures made? Using what software? Or taken from where?

Response 02: The structures were made using ChemDraw software. The information has also been stated in the manuscript.

Comment 03: In the figure legend or in the figure if possible for figure 10 and 11 show all of the bioactive ingredients that interfere with the major parts of these pathways.

Response 03: All the bioactive ingredients that interfere with the major parts of these pathways have been included in the figures mentioned above.

Comment 04: This review is an excellent summary of the literature but it seems so descriptive. I believe a review should add analysis/interpretation and direction for the field. This is only a summary of journal articles.

Response 04: A critical and collective discussion as well as analysis/interpretations and future directions have been added to the review under a new section namely “Reported Cytotoxic Constituents: Therapeutic Perspective and Future Directions”.

Comment 05: How were these articles chosen? How many exist versus how many were summarized here?

Response 05: A methodological section has been added to the review with description of the databases used, search key words as well as the inclusion and exclusion criteria. The authors have tried hard to include all the articles they’ve found (which met the inclusion criteria).

Round 2

Reviewer 2 Report

The authors appear to have made an effort to comply with the critical issues that were raised. In particular, they integrated their review with data resulting from a survey of the database of the Developmental Therapeutic Program of the National Cancer Institute, USA, and with some pharmacokinetic data, whenever available. This has contributed to make of their review something more than a list of plant genera and species. Also, they have outlined better the limitations and the introductory value of their survey. The two partially overlapping Tables have been combined into one. Their manuscript, although still of low-moderate interest, may now be considered for publication in Molecules.

I still recommend, though, that the text be subjected to a further linguistic revision for English vocabulary, syntax and style. There are still numerous wording and spelling mistakes, also in systematic names (see, as an instance, Luffa cylindrical, instead of Luffa cylindrica, and Phagnalon rupstre for Phagnalon rupestre).

I also recommend that the contents of column 4 of Table 1 (Study method) be summarized, as they occupy too much space at present. Also the heading of this column should be changed to "Study methods".

Author Response

Dear Reviewer,

Thank you again for your kind review and further suggestions. Please find the response below:

Suggestion 1: I still recommend, though, that the text be subjected to a further linguistic revision for English vocabulary, syntax and style. There are still numerous wording and spelling mistakes, also in systematic names (see, as an instance, Luffa cylindrical, instead of Luffa cylindrica, and Phagnalon rupstre for Phagnalon rupestre).

Response 1: A further linguistic revision for English vocabulary, syntax and style have been carried out to the best of the Authors’ knowledge. The wording and spelling mistakes were corrected by using the “Spelling and Grammar” checking tool. Again, all the botanical names were checked again and the incorrect ones were corrected. All the changes were tracked so that they can easily be identified.

Suggestion 2: I also recommend that the contents of column 4 of Table 1 (Study method) be summarized, as they occupy too much space at present. Also the heading of this column should be changed to "Study methods".

Response 2: The heading of column 4 of table 1 was changed to “Study Methods”. The contents of column 4 were also summarized in such a way that they occupy less space.

Reviewer 3 Report

Thank you for making the requested changes.

Author Response

Dear Reviewer,

Thank you again for your kind suggestion, response and support.